# An *Intranet* of Things approach for adaptable control of behavioral and navigation-based experiments

John C Bowler[1,2,3]*[†], George Zakka[1,2]*[†], Hyun Choong Yong[1,2], Wenke Li[4], Bovey Rao[1,2,5], Zhenrui Liao[1,2], James B Priestley[6], Attila Losonczy[1,2]*

[1]Department of Neuroscience, Columbia University, New York, United States; [2]Mortimer B. Zuckerman Mind Brain Behavior Institute, Columbia University, New York, United States; [3]Department of Neurobiology, University of Utah, Salt Lake City, United States; [4]Aquabyte, San Francisco, United States; [5]Doctoral Program in Neurobiology and Behavior, Columbia University, New York, United States; [6]Brain Mind Institute, École polytechnique fédérale de Lausanne, Lausanne, Switzerland

*For correspondence:
jack.bowler@utah.edu (JCB);
gz2333@columbia.edu (GZ);
al2856@cumc.columbia.edu (AL)

[†]These authors contributed equally to this work

Competing interest: The authors declare that no competing interests exist.

## eLife Assessment

Bowler et al. present a software/hardware system for behavioral control of navigation-based virtual reality experiments, particularly suited for pairing with 2-photon imaging but applicable to a variety of techniques. This system represents a **valuable** contribution to the field of behavioral and systems neuroscience, as it provides a standardized, easy to implement, and flexible system that could be adopted across multiple laboratories. The authors provide **compelling** evidence of the functionality of their system by reporting benchmark tests and demonstrating hippocampal activity patterns consistent with standards in the field. This work will be of interest to systems neuroscientists looking to integrate flexible head-fixed behavioral control with neural data acquisition.

**Abstract** Investigators conducting behavioral experiments often need precise control over the timing of the delivery of stimuli to subjects and to collect precise times of subsequent behavioral responses. Furthermore, investigators want fine-tuned control over how various multi-modal cues are presented. behaviorMate takes an 'Intranet of Things' approach, using a networked system of hardware and software components for achieving these goals. The system outputs a file with integrated timestamp–event pairs that investigators can then format and process using their own analysis pipelines. We present an overview of the electronic components and GUI application that make up behaviorMate as well as mechanical designs for compatible experimental rigs to provide the reader with the ability to set up their own system. A wide variety of paradigms are supported, including goal-oriented learning, random foraging, and context switching. We demonstrate behaviorMate's utility and reliability with a range of use cases from several published studies and benchmark tests. Finally, we present experimental validation demonstrating different modalities of hippocampal place field studies. Both treadmill with burlap belt and virtual reality with running wheel paradigms were performed to confirm the efficacy and flexibility of the approach. Previous solutions rely on proprietary systems that may have large upfront costs or present frameworks that require customized software to be developed. behaviorMate uses open-source software and a flexible configuration system to mitigate both concerns. behaviorMate has a proven record for head-fixed imaging experiments and could be easily adopted for task control in a variety of experimental situations.

## Introduction

In this work, we present *behaviorMate*, an open-source system of modular components that communicate with each other over a local area network (LAN), orchestrated by a custom Java application. We validate our approach through benchmarking and experimental results, demonstrating the ability of behaviorMate to control head-fixed navigational experiments both in a self-powered treadmill (TM) and in a fully visual virtual reality (VR) setup with a running wheel. This solution permits researchers to select the configuration of cues they wish to present at run-time, resulting in a system that is capable of adapting quickly to incorporate changes to experimental protocols. The modular design makes it easy to adapt to shifting demands of research on the same behavioral setups while producing a consistent, easy-to-parse output file describing the experiment.

Proprietary systems such as National Instrument's Data Acquisition Systems (DAQ) and their accompanying software controller, LabView, typically have large upfront and licensing costs for each experimental rig their system is deployed on. behaviorMate exclusively uses open-source software and is simple to construct with parts that are significantly cheaper. The customized electronics may be ordered as printed circuit boards (PCBs) and assembled by hand or purchased preassembled. The latter approach saves assembly time while adding significant expense; however, this can be mitigated by bulk ordering. The total cost of one of VR rig including five computer monitors, Android computers to render the visual scene, a frame for holding the monitors, and a lightweight running wheel is approximately $3700. The total cost for a belt-based TM system is around $1700. For two-photon (2p) imaging, additional components may be needed, such as a light-tight and laser resistant enclosure, which were not factored into the calculation. While other open-source and modular behavioral control systems have been developed which permit a variety of use cases (*Akam et al., 2022*; *Saunders et al., 2022*), behaviorMate does not require the user to write software code and uses compiled Arduino programs that do not incur the overhead of code interpreters to maximize performance.

Many experiments rely on using head-restrained animals to study the neuronal circuits underlying complex behaviors; such immobilization poses a fundamental challenge to behavioral research as it stymies animals' natural movement through their environments. In order to address this issue, a plethora of 'treadmill' systems have been developed relying on belt-based systems (*Royer et al., 2012*; *Lovett-Barron et al., 2014*; *Jordan et al., 2021*), VR (*Dombeck et al., 2010*), or augmented reality (AR), where experimenters move 'spatial' cues to create the illusion of locomotion without actual movement in physical space (*Jayakumar et al., 2019*). Belt-based systems involve placing an animal on a physical belt TM where, as the animal runs, physical cues affixed to the belt provide spatial information. VR and AR systems operate under closed-loop control, where the environment responds to the animal's behavior (*Dombeck and Reiser, 2012*). Normally, these cues are visually displayed on one or more computer monitors, but olfactory and auditory stimuli have also been utilized at regular virtual distance intervals to enrich the experience with more salient cues (*Radvansky and Dombeck, 2018*; *Fischler-Ruiz et al., 2021*). Furthermore, belt- and VR/AR-based elements have been combined to enrich experiences and enable more complex behavioral tasks. At times, traditional 'open-loop' stimulus may be required, such as timed cue presentations, allowing pre- and post-event neuronal activity to be examined. Increasingly, studies combine open- and closed-loop behavioral paradigms to identify stimulus–response profiles or behavioral state driven changes to stimulus responses. The proliferation of various experimental paradigms demands a high level of flexibility in experimental setups, so that methods can be integrated within single experiments or across lines of research to interrogate neuronal function.

Many of the spatial representations observed in freely moving animals are conserved in head-restrained animal setups (*Aronov and Tank, 2014*; *Dombeck et al., 2010*). Observations within the medial temporal lobe, focused on all subregions of the hippocampus as well as the entorhinal cortex, have shown that prominent neural correlates of navigation behaviors exist in both the TM belt style systems as well as in the visual only VR systems (*Aronov and Tank, 2014*; *Dombeck et al., 2010*; *Royer et al., 2012*; *Danielson et al., 2016*). In CA1 (*Danielson et al., 2017*; *Zaremba et al., 2017*) and CA3 (*Terada et al., 2022*), place cells, goal-related tuning, and other feature-selective tuning have been observed. Additionally, in visual VR systems, grid cell-like tuning has been observed in the medial entorhinal cortex, as was distance tuning (*Aronov and Tank, 2014*; *Heys et al., 2014*). The increased control over reward distribution and cue manipulation that VR systems afford has been pivotal for recent findings relating to how cells in CA1 form place fields, uncovering novel mechanisms

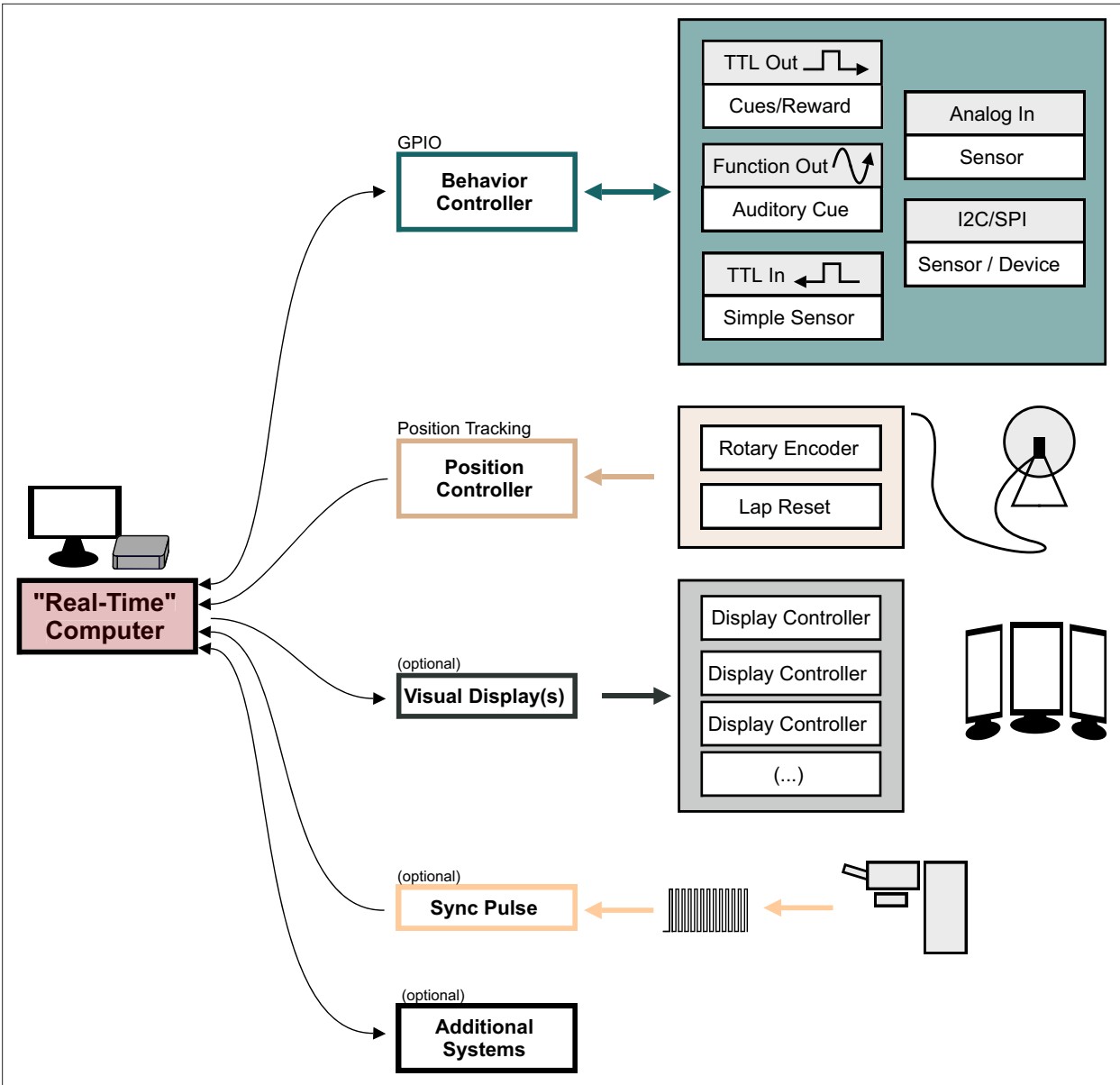

**Figure 1.** Overview of the behaviorMate system with optional components. Diagram shows the modular components behaviorMate can interact with the colored arrows show the direction of information flow. For example, the Position Controller module only receives position updates and forwards them to the computer. The Visual Display module sends data to the Display Controllers to render the scene. The Behavior Controller performs a general-purpose input/output (GPIO) function and may both send and receive data to and from the Computer. An optional external Sync Pulse is shown to demonstrate that behaviorMate will log any UDP received UDP packets, such as timestamped sync-signals which could be beneficial in certain setups to synchronize neural data with behavior. Additional Systems may also be implemented on the behaviorMate intranet, taking advantage of the flexible UDP messaging scheme.

underlying synaptic plasticity (*Gonzalez et al., 2023*; *Bittner et al., 2017*; *Priestley et al., 2022*), demonstrating the power of these systems to test specific theoretical predictions.

## Results

behaviorMate is an integrated system composed of several sub-components that communicate on an LAN using JSON-formatted packets transmitted via a standard Datagram packet protocol (UDP) (*Figure 1*). This approach allows for modularity and encapsulation of different concerns. Moreover, acknowledgment packets are used to ensure reliable message delivery to and from connected

hardware. We dub the resulting system as an '*Intranet* of Things' approach, mimicking the Internet of Things systems commonly found in modern 'smart homes', which communicate exclusively on a dedicated LAN using statically assigned internet protocol (IP) addresses. This allows for a central user interface (UI) or 'real-time' computer to seamlessly send and receive messages from various accessory downstream devices that have dedicated simple functions. Importantly, the techniques we describe are possible implementations of the behaviorMate system. In our experimental setups, we use behaviorMate to combine in vivo head-fixed mouse navigation experiments with 2p microscopy, focusing on one-dimensional (1D) spatial navigation tasks (Figure 3). Due to its modular design, however, behaviorMate can be easily reconfigured to provide closed- or open-loop control during a variety of behavioral paradigms (and has been used for non-navigation-based studies; such as the head-fixed trace fear conditioning task described in *Ahmed et al., 2020*).

In addition to running a PC-based UI, we combine Arduino open-source micro-controllers with custom designed PCBs to run the experiments (*Figures 1 and 2D, E*, *also see* Appendix 1). The circuits include Ethernet adapters, allowing the Arduinos to communicate on the network, as well as connectors that interface them to sensors or actuators for controlling the experiments or for reporting the behavioral state back to the UI. We outline a general-purpose input/output (GPIO) circuit that can connect to a variety of sensors and reward valves generally using a transistor–transistor logic (TTL) pulse/square wave as well as I2C and serial peripheral interface (SPI) protocols (*Figure 2D*), a position tracking circuit (*Figure 2E*) that is dedicated mainly to reading updates of the animal's location from a rotary encoder, and a VR controller setup which can be used to provide a more immersive visual display to animals during the behaviors. Importantly, these individual components can be swapped out or added to with increased experimental demands. In most cases, the provided software and hardware are sufficient to meet users needs, however, we do also point to certain targets for easy expansion which would require editing the software provided to interface with specialized hardware (e.g. adding a novel SPI-based extension to the GPIO circuit). In general, without editing software the user has two options for controlling hardware: using one of the digital/analog port on the GPIO board or over network via an Ethernet adapter (but this could require following the same JSON API as existing behaviorMate components). For additional integration of software components or testing, the UI can be configured to send messages to the PC's localhost address (i.e. 127.0.0.1 on most PCs). To speed up development, we created virtual implementations of the electronics in Python code to test UI changes before testing them on a physical system. Most modern programming languages, including Python and Matlab, have built-in support for JSON processing and UDP networking making component and new feature testing easy with the help of utility scripts that can be written in users' preferred environments (*see* Appendix 1 for references to example Python code).

Lastly, since our main focus is on 1D navigational tasks for 2p imaging, we provide two designs for physical rigs, which hold animals fixed in place under a microscope. However, they could also be adapted to any other methods for in vivo recording. One system is a self-powered, belt-based TM system, wherein animals are placed on a fabric belt that they pull beneath them to advance their virtual position. Locations and contexts can be distinguished in this setup by having various fabrics and physical cues. Alternatively, the other system provides navigational contexts through a series of computer displays for a purely visual experience, but allowing for additional flexibility of within-task manipulations to context and cues. We describe the details of both of these systems and, additionally, note that with the following methods, it is possible to combine aspects from both belt-based and VR systems to meet novel experimental designs. To clarify, for the remainder of the paper, 'treadmill' will refer to these belt-based systems while 'VR' will refer to the visual-based system in which animals moving a plastic running wheel.

## The user interface

The central hub for running experiments is the behaviorMate UI (*Figure 2A*). The UI collects input from various sensors, assembling information about virtual position and behavior state data asynchronously. The UI performs three main functions: (1) accepting user input and displaying the current state of experiments to ensure proper functioning, (2) writing hardware information in a timestamped streaming text file which can be processed later for analysis of the experiment, and (3) serving as the central communications hub for sending and receiving UDP-formatted messages to the downstream components. The system is event-driven such that the hardware components send messages to the

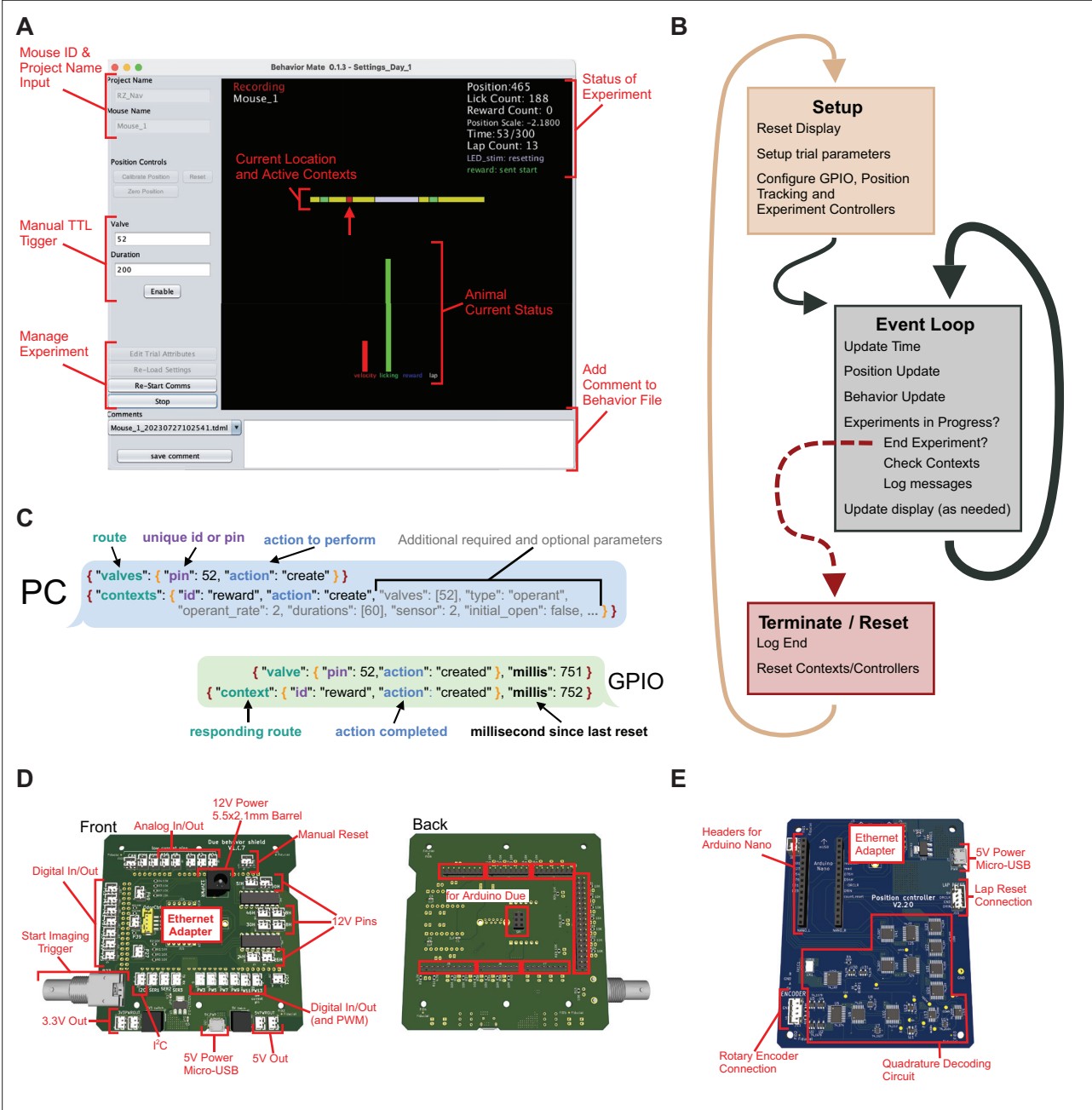

**Figure 2.** Details of user interface (UI) function. (**A**) Screenshot of the UI. The interface provides a snapshot of the animal's current status (*center*) as well as cumulative information about the current experiment (*upper-left*). The control panel along the left side provide: *Top*, an input for a project name and mouse id, which control where the resulting behavior file will be saved (these boxes are disabled when an experiment is currently running); *Middle*, controls to trigger the general-purpose input/output (GPIO) to send a transistor–transistor logic (TTL) pulse that is to issue a water reward or turn on an LED; and *Bottom*, controls to start/stop experiments as well as to load settings files. (**B**) Details of UI event loop. The UI is continuously executing an event loop on every update step which checks for messages, writes the behavior file and updates the display. (**C**) JSON-formatted message passing. The PC sends and receives JSON-formatted messages via UDP to active components of the system. Messages have a nested structure to allow for internal routing and subsequent actions to be taken. (**D**) Rendering of the GPIO circuit used. *Left*: JST headers connected to I/O pins and pull-down resisters to allow for easy connections with peripheral devices that can be activated by TTL pulses (both at 3.3 and 12 V, depending on the pins used). Additionally, power connections and a BNC output trigger are provided. *Right*: This board attaches to an Arduino Due microcontroller. (**E**) Updates to the position have a dedicated circuit for decoding quadrature information and sending updates to the PC. Connections for a quadrature-based rotary encoder, power, and Ethernet are provided. Additionally, an input for a 'Lap Reset' sensor is provided and headers to interface with an Arduino Mini.

UI when the state changes rather than the UI continuously polling. There are currently two versions of the UI. The first is a long-term support release that evolved alongside the experimental systems and has been used for data collection for numerous published studies; the second is a streamlined beta version that has been rewritten to take advantage of the JavaFX visual development toolbox. Java was chosen for the UI because it is cross-platform in that it can be run on Windows, Linux, and Mac, and supports standard networking protocols. For consistency, interchangeability, and space optimization, we run the UI on an Intel(TM) NUC mini PC with a 2.4-GHz base-frequency (up to 4.2 GHz based on processor load) 4-core i5-1135G7 processor, 16-GB DDR4-3200MHz RAM, and a 512-GB NVMe M.2 SSD hard drive running Window 10 Professional Edition, although many other configurations are supported. Notably, due to the system's modular architecture, different UIs could be implemented in any programming language and swapped in without impacting the rest of the system. The only requirement for interchangeability of the UI is to implement sending and receiving of the same JSON-formatted UDP messages packets to the downstream hardware components (*see* Appendix 1 for Arduino firmware which includes additional detail of the JSON messaging protocol).

The UI executes in three distinct phases during run-time. First, a setup routine is executed, then the program enters the main event loop, and finally a termination routine is executed when an experiment is completed (*Figure 2B*). The main event loop begins executing as soon as the UI is loaded, but additional steps for checking and logging the contexts' state are added during the experiment. Devices still receive input and deliver output to the UI even when an experiment is not currently running, allowing experimenters to confirm that everything is working before starting. Experiments using behaviorMate are configured by user-generated JSON files with key–value pairs specified in the documentation. These settings files can be manually generated in a text editor or programmatically generated using a script (*see also* Appendix 1).

The main structure of the application logic is an event loop that iterates over enabled *contexts*. Within behaviorMate, a context is grouping of one or more stimuli that get activated concurrently. For many experiments, it is desirable to have multiple contexts that are triggered at various locations and times in order to construct distinct or novel environments. Contexts define presentation schemes for both simple stimuli such as an LED lighting up and water reward delivery, as well as more complex stimuli like flashing LEDs or even an entire visual VR scene. A context can be applied globally or to one or several locations; a list of locations and rules governing an individual context's activation is referred to as a Context List. When a Context List is active, the devices assigned to it will be triggered. For example, a user could configure an LED which blinks at several designated locations along the track. Context Lists are implemented in the UI using an Object-Oriented approach that streamlines implementing logic for governing the presentation and control of novel cue types. The main rules governing how a Context List works are implemented in a check method which is passed the current state of the experiment as input arguments and returns a Boolean variable as true or false corresponding to whether a Context List should be active or not. In this way, novel Context List types may be added to the program easily through inheritance of a base class and overriding this one method. Additionally, Context Lists are instantiated using the 'Factory Method' pattern (*Gamma and Helm, 1994*) and appended to a dynamic list. This pattern allows for adding complex functionality while minimizing the impact on performance and increasing code reuse and modularity. Existing Context List types can be modified by applying 'decorators' (*Gamma and Helm, 1994*), resulting in *composable* behavioral paradigms, meaning that novel experiments can be implemented by nesting Context Lists within one or more existing decorators, without requiring any software modification. Settings files following JSON syntax specify the configuration as well as the context lists and decorators. For example, a particular decorator can be configured to cause a context to only be active on even laps while a second decorator may restrict the context from being active until after a certain amount of time has passed. Composing these two rules means it is possible to trigger the contexts on even laps, only after 5 min have passed since starting the experiment. However, if the available set of decorators is not enough to implement the required task logic, some modifications to the source code may be necessary. These modifications, in most cases, would be very simple and only a basic understanding of object-oriented programming is required. A case where this might be needed would be performing a novel customized real-time analysis on behavior data and activating a stimulus based on the result. The JavaFX version of the behaviorMate UI additionally supports a plugin architecture which will further simplify the process of adding custom decorators and context list rules in the future.

## Behavior tracking and control

### GPIO/Behavior Controller

The primary GPIO circuit we implement, referred to as the Behavior Controller, is composed of an Arduino attached to a custom circuit board (*Figure 2D*). The PCB handles voltage regulation and provides the necessary connectors to operate downstream devices. The Arduino program distributed with behaviorMate controls all connected devices and communication between them and the UI. The program wraps 'action' messages with 'routes' (*Figure 2C*). This pattern of routing messages simplifies debugging since it is clear which classes of the Arduino code received the messages and where replies were generated. The messages are JSON-formatted text, so they are human-readable and can be simulated within the behaviorMate PC for testing and debugging purposes (*also see* Appendix 1, for links to Arduino firmware and documentation on the JSON messaging protocol). The PCB has various connectors and other components that makes it easier to connect arbitrary hardware, including sensors and actuators, to the Arduino. Sensors and actuators can be connected to the controller using one of the 13 digital or 5 analog input/output connectors. Moreover, the controller provides an Ethernet adapter for the Arduino to permit LAN communication.

The Behavior Controller has several important functions; it receives signals from the UI to activate rewards and cues such as odor valves, LEDs, and tone generators and passes input from sensors to the PC. In our setup for 2p imaging, the Behavior Controller also sends a synchronizing signal to the microscope to initiate imaging after an experiment is started through the behaviorMate UI. Some microscopes can be configured to begin imaging by a TTL synchronizing trigger. Alternatively, micro-scopes might send a trigger indicating that recording has commenced. Moreover, some setups can periodically send back synchronization signals to ensure alignment during long recordings that may be affected by clock drift. The Behavior Controller can both receive and send triggers for the purpose of synchronization which will be timestamped and logged by the UI. By default, this signal is a TTL pulse (3.3 V in our implementation, *Figure 2D also see* Appendix 1) and can also be configured to activate other types of recording devices such as video cameras or electrophysiology interfaces. However, it is important to note that the sampling rate of an Arduino Due (used in Behavior Controller) is incapable of the kilohertz sampling rate of typical electrophysiology devices. In these cases, it might be neces-sary to send a periodic synchronization TTL from the behavior controller to the electrophysiology hardware to ensure proper timing of behavior and neural data. behaviorMate is compatible with a variety of alignment and synchronization schemes and it is left up to the users to implement the solu-tion most appropriate for their experimental setups.

### Position controller

The position controller's function is to detect animal movement and report it to behaviorMate. It is composed of an Arduino Nano attached to a custom circuit which has an onboard quadrature decoder, 16-bit counter, and connectors for attaching the rotary encoder and lap-reset circuit. For either TM or running wheel-based setups (i.e. for a VR setup), a quadrature rotary encoder device is coupled to the shaft of a wheel that turns as the animal runs. The turning of the rotary encoder generates necessary signals, or 'ticks', which are passed to the quadrature decoder to calculate the instantaneous velocity of the animals. These quadrature ticks are decoded and counted in the counter. The custom circuit (*Figure 2E*) uses a 10-MHz oscillator which is capable of counting high resolution encoders (>4096 ticks/turn) at greater than 1K RPM. The onboard counter enables the tick counting to be decoupled from the speed of the on-board Arduino. Consequently, the Arduino can check the counter at a much slower rate (100 Hz in our setup, but can be as slow as 5 Hz) without losing count of the ticks. The Arduino is only responsible for checking the counter and generating a JSON-formatted text string to transmit to the behavior PC via Ethernet, and only transmits when there is movement. Significantly, this eliminates polling from the behavior PC and allows the communication between the Arduino and the PC to be one way instead of bidirectional. The behavior PC then translates the counts into a meaningful measurement of the linear distance traveled by the animal on the belt or running wheel.

## Behavioral apparatus

### TM system

Self-powered TM systems have been extensively used in the study of navigational behaviors (*Royer et al., 2012*; *Tuncdemir et al., 2022*; *Lovett-Barron et al., 2014*; *Grienberger and Magee, 2022*;

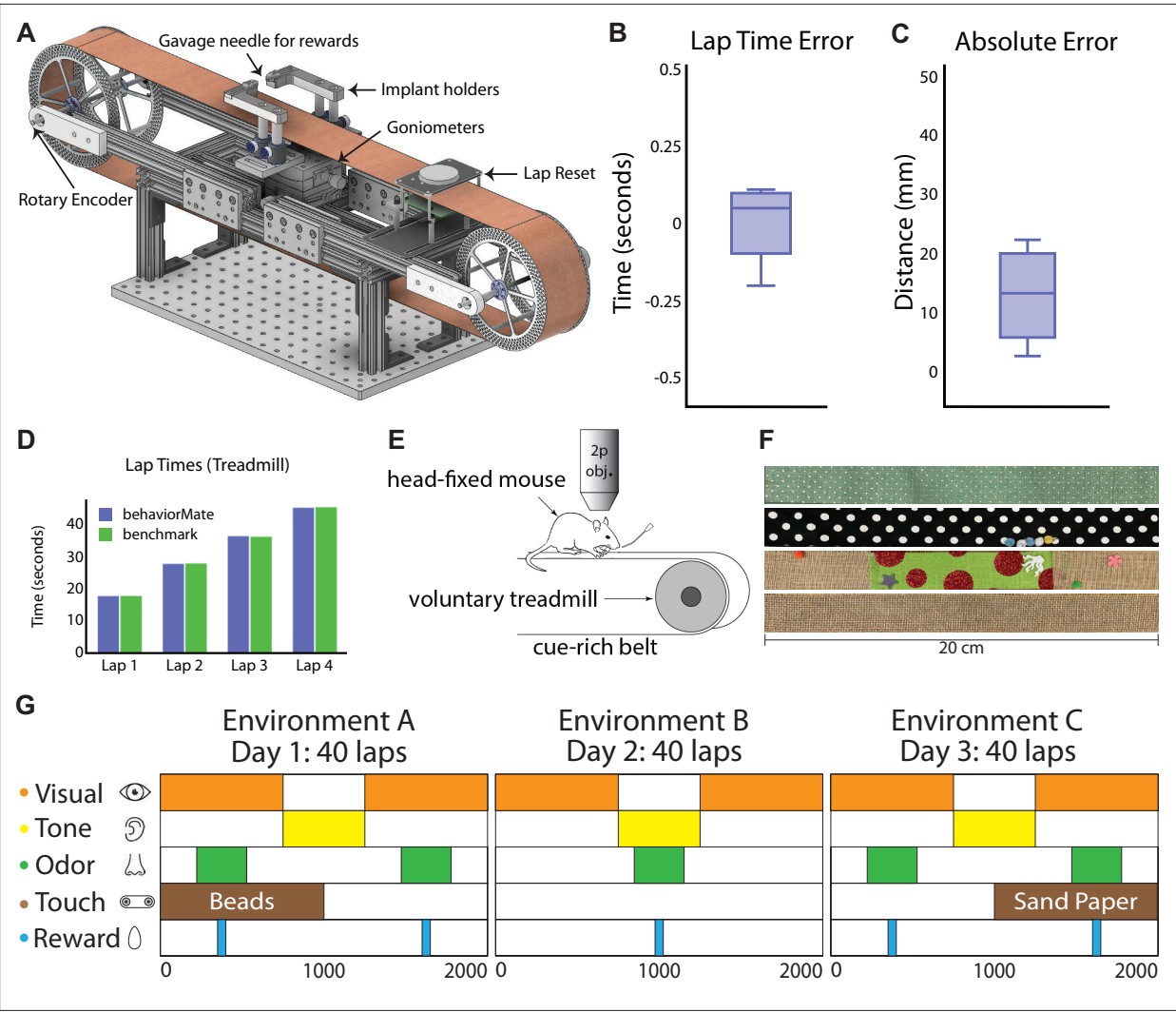

**Figure 3.** Details of treadmill system. (**A**) CAD model of voluntary treadmill that can expand or contract to accommodate different sized running belts. Implant holders sit on goniometers to allow mouse head angle to be adjusted. (**B**) Lap time error is defined for each lap (full turn of the treadmill belt) as the difference between the time recorded by behaviorMate and the computer vision benchmark in second ($\text{time error} = 0.0025 \pm 0.1128\,\text{s}$, mean $\pm$ std). (**C**) Absolute error is defined for each lap as the difference between the position in behaviorMate and the benchmark position in millimeters ($\text{position error} = 13 \pm 7.6581\,\text{mm}$, mean $\pm$ std). (**D**) The time to complete a lap according to behaviorMate and the benchmark were nearly identical. (**E**) Schematic of head-fixed behavioral task. Two-photon objective. (**F**) Cue-rich belts for treadmill behavioral tasks. Belts are easily interchangeable. (**G**) Example test condition spanning 3 days and involving five sensory cues. Each row represents a sensory modality (visual, auditory, etc.). Cues are tied to locations on the belt or wheel as indicated by the colored rectangles. Locations and durations of cue presentations can be changed across trials.

*Geiller et al., 2017*). The following section outlines a protocol for the construction of a behaviorMate compatible mouse TM setup that has been used for head-fixed imaging (*Kaufman et al., 2020*; *Terada et al., 2022*; *Rolotti et al., 2022b*). In combination with the software and circuits described above, the TM system provides a flexible platform for implementing novel experimental paradigms.

## Frame

The TM system requires a frame to support it and maintain wheel alignment, so we provide designs for a frame made of extruded aluminum rails (*Figure 3A*). The rails support wheels on either end for a fabric 'belt' that can be moved by a mouse. In addition to the wheels, we added two wheel-rollers that support the belt near the animal's fixed position (*Figure 3E*). This design does not have a platform directly beneath the mouse, allowing the belt to absorb vibrations, which reduces motion artifacts and allows for a stable field of view during in vivo imaging experiments. TM belts are constructed from 2-inch-wide fabric ribbon and may be embellished by linking multiple fabrics or adding various

physical cues such as foam balls, sequins, or hook-and-loop fasteners for texture (*Figure 3F*). These cues help the animal to orient itself and are useful for information encoding. In this setup, the animal is head-fixed beneath a microscope, but is able to move the belt and attached physical cues which mimics navigation while still allowing investigators to perform imaging of cellular and even subcellular regions of interest (ROIs). To further optimize the TM for imaging, a two-goniometer device placed underneath the belt is used to adjust the angle of the animal's head along the roll (side-to-side) and pitch (forward and backward) axes. This setup allows the investigator to level the field of view, ensuring the neuronal plane of interest is perpendicular to the optical axis of the microscope, thus increasing the number of neurons that can be imaged and improving image clarity. However, this may not be needed in setups where the objective itself can be tilted; if the objective can be tilted along two axes, no goniometers would be needed, whereas if the objective can only tilt along one axis, a single goniometer could be employed. Finally, the frame has a sliding mechanism to adjust the distance between the wheels and accommodate various sized belts to ensure proper tension. Proper belt tension ensures the animal can move easily and reduces the chance of belt slippage along the wheels.

## Electronics and peripherals

Interfacing between the TM and behaviorMate requires the previously described electronics. Any number of optional peripheral devices such as LEDs and odor release systems may be added. A rotary encoder is attached to one of the two wheel shafts and connected to the position tracking circuit to log the animal's movement. The behaviorMate UI handles the conversion of the rotary encoder's spinning to the linear distance traveled. It also maintains the animal's current lap number. An additional 'lap-reset' feature prevents drift from accumulating between the digital tracking and physical TM belt. We provide designs for an optical lap-reset circuit for determining when a complete revolution of the belt has been made (*Figure 3A*). Setting up a lap-reset circuit additionally permits a calibration function that can be temporarily enabled prior to running an experiment to adjust the scaling between rotary encoder 'ticks' and distance moved along the TM belt.

Generally, position tracking drift is small (*Figure 3C*) but can occur (especially with more 'slippery fabrics' or if belt tension is set too low). It is important to accurately specify the track length and calibrate the position scale factor to tightly couple the PC and the electronics. If the system is not set up properly or the belt is not sufficiently tight, the belt may slip, leading to inaccurate position readings. Therefore, we provide a benchmark of the position tracking performance under normal conditions. To compute the linear distance traveled, a small LED was attached to the top of the belt and the entire TM system was moved to a completely dark room with an IR camera pointed directly at the belt. The belt was moved by hand and upon each complete revolution, the LED would pass within the field of view of the camera. OpenCV (*Bradski, 2000*) was used to extract the lap completion times from the recorded video. These were taken to be the 'ground truth' lap completion times. The quantities of interest were the time and position differences between the LED and those tracked by the behaviorMate UI. The mean of the magnitude of the differences between lap times reported and the 'ground truth' was 0.102 s (*Figure 3B, D*), while the mean difference between position upon lap completion was 13 mm and did not exceed 23 mm (*Figure 3C*). Given that the mean and max differences were a fraction of the average size of a subject mouse and significantly smaller than the typical size of reward zones, these metrics were deemed more than acceptable. Enabling the lap-reset feature on self-powered TM prevents accumulation of these errors, ensuring that position, as tracked by the UI, is aligned with the fabric belt and cue presentations are aligned with the positions tracked by the UI.

The UI synchronizes any number of additional sensory cues such as LEDs, tones, and odor release systems, which can be present at pre-defined times or locations along the belt (*Figure 3G*). For example, a liquid reward can be coupled to a specific location on the TM belt that is either *operant* (delivered only if the animal licks within a 'reward zone') or *non-operant* (delivered immediately when the animal reaches a 'reward zone'). These 'reward zones' are defined by a location and radius, so there is user-specified tolerance for the areas animals can receive operant rewards. The rules governing the rewards are fully customizable and specified in the settings file for the UI.

## Use in past studies

The physical design of the TM was inspired by *Royer et al., 2012* and influenced by several other hippocampus studies focusing on behavioral navigation tasks (*Kaifosh et al., 2013*; *Lovett-Barron et al., 2014*). Our design is flexible and has been validated by the myriad experimental paradigms that have been successfully implemented our lab (*Tuncdemir et al., 2023*; *Vancura et al., 2023*; *O'Hare et al., 2022*; *Rolotti et al., 2022a*; *Rolotti et al., 2022a*; *Terada et al., 2022*; *Geiller et al., 2022*; *Blockus et al., 2021*; *Grosmark et al., 2021*; *Geiller et al., 2020*; *Kaufman et al., 2020*; *Turi et al., 2019*; *Zaremba et al., 2017*; *Danielson et al., 2017*; *Danielson et al., 2016*) as well as set up and run by collaborators (*Tuncdemir et al., 2022*; *Dudok et al., 2021a*; *Dudok et al., 2021b*; *Rolotti et al., 2022a*). Multi-modal cues can be incorporated across several days of experiments to probe different aspects of navigation and associative learning (*Figure 3G*). For example, the TM has been used to test goal-oriented learning, where goal locations are 'hidden' and animals must report their understanding of the correct location by licking at the reward spout in order to trigger water delivery (*Zaremba et al., 2017*; *Danielson et al., 2016*). During these tasks, novel cues can be presented by introducing new TM belts as well as novel non-spatial cues such as background tones, blinking LEDs, or odor stimuli. Moving the reward location forces animals to learn updates to task rules mid-experiment and has been used to investigate deficits in mouse models of psychiatric disorder (*Zaremba et al., 2017*) as well as neural circuit analysis of the mechanisms behind reward learning (*Kaufman et al., 2020*) and the development of signals underpinning spatial navigation (*Rolotti et al., 2022b*; *Terada et al., 2022*; *Danielson et al., 2016*). Cues can be tied to time, location, or both. Switching between a time-dependent presentation and a location-dependent one provides a powerful tool for assessing how the same cue may be processed differently based on the current context, location, or task (*Terada et al., 2022*; *Tuncdemir et al., 2022*). In addition to cue presentation, the setup has also been used to trigger optogenetic stimulation (*Kaufman et al., 2020*; *Geiller et al., 2022*; *O'Hare et al., 2022*; *Rolotti et al., 2022b*); this coupling of navigational cues provided by the TM belt with stimulation in hippocampal region CA1 was used for the first successful optical induction of place fields (*Rolotti et al., 2022b*).

## VR system

The use of VR cues can add significant flexibility to behavioral studies and has thus become increasingly popular in neuroscience. Although different VR implementations have been described, for this manuscript, VR is defined as closed-loop control from the behaviorMate UI that integrates position updates virtually and advances cues past the animal to create the illusion of movement, unlike a belt which physically moves past the animal. Various systems for accomplishing this have been proposed and implemented previously (*Dombeck et al., 2010*; *Harvey et al., 2012*; *Sheffield et al., 2017*; *Hainmueller and Bartos, 2018*; *Campbell et al., 2018*; *Arriaga and Han, 2017*; *Arriaga and Han, 2019*; *Dudok et al., 2021a*). Mice can learn to use VR cues as landmarks for traversing environments to specific locations. Neuronal circuits thought to be involved in navigation in freely moving mice have been shown to be similarly engaged during the described VR tasks (*Dombeck and Reiser, 2012*; *Dombeck et al., 2010*; *Aronov and Tank, 2014*). behaviorMate represents a novel contribution in that it incorporates visual VR displays into its architecture for 1D navigation and is designed to adapt seamlessly to novel arrangements of VR displays with minimal computational load for each additional display (*Priestley et al., 2022*; *Bowler and Losonczy, 2023*). This is achieved by rendering the virtual environments on separate Android devices (one per display) with a separate application (VRMate). Since the same behaviorMate UI is used for TM and VR experiments, all of the experimental paradigms described in the TM section can be implemented in the VR system with minimal changes to the settings file. Additionally, VR permits rapid visual context switches and the addition of novel virtual objects in a familiar scene. Furthermore, 'hybrid' setups are also supported in which visual displays can be added to physical TMs in order to combine both tactile and visual cues.

## VRMate

VRMate is the companion program to behaviorMate that runs the visual VR interface to display cues to animals as they move in virtual space. VRMate is a program developed using the Unity(TM) game engine and written in C#; it listens for incoming UDP packets carrying position updates from the behaviorMate UI. In our implementation, each visual display is connected to an ODROID C4 which

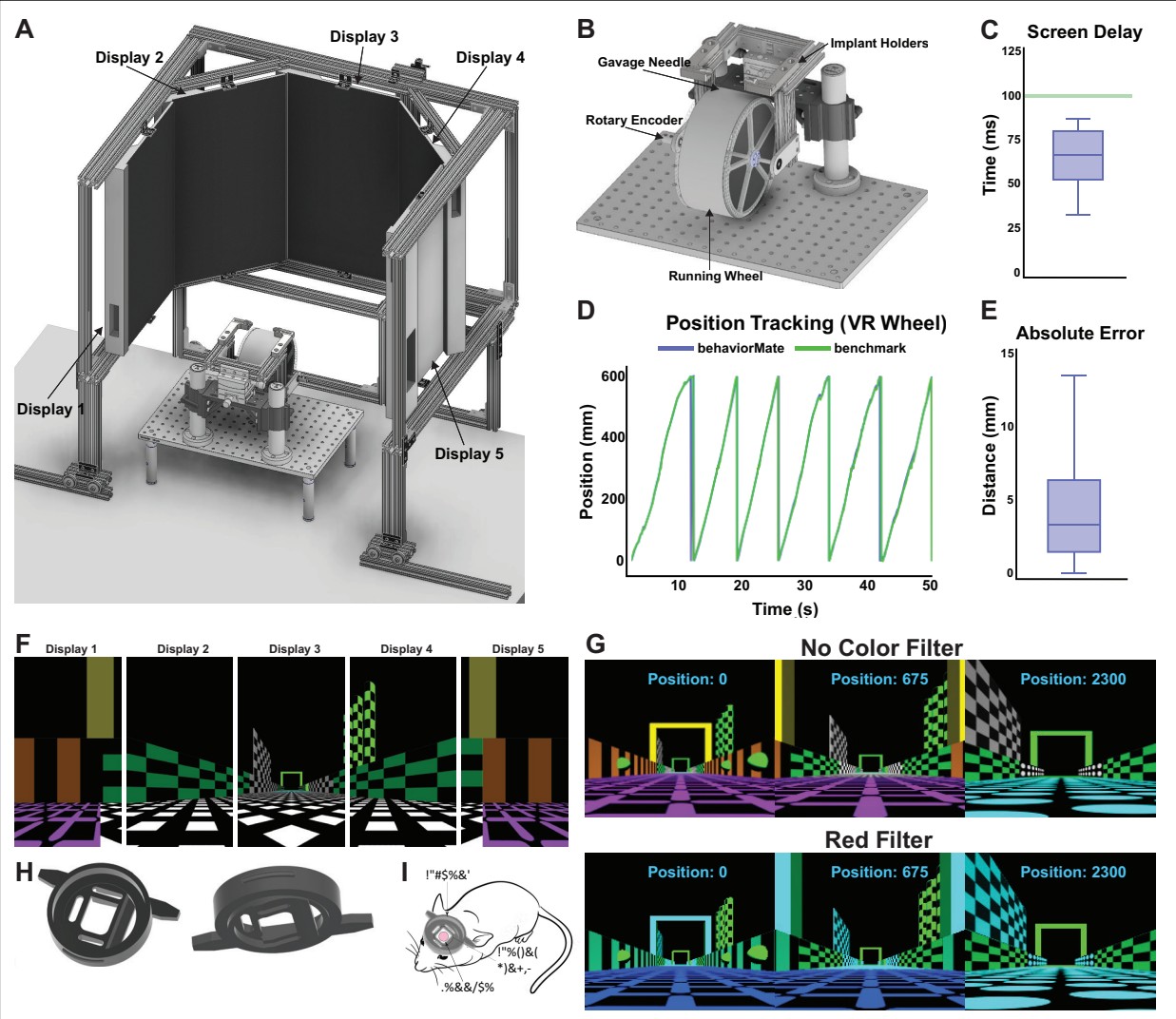

**Figure 4.** Details of virtual reality (VR) system. (**A**) The complete assembly. Five monitors are suspended in an octagonal pattern in front of the subject and display the virtual scenes. (**B**) A running wheel attached to a goniometer to allow for angle of subject's head to be adjusted. Placed directly underneath the objective. (**C**) Box plot describing the delay between when the running wheel is moved and when the virtual scene is updated to reflect the new position (delay $= 0.0660 \pm 0.01570$ s, mean $\pm$ std). (**D**) Plot showing the difference between the position of the animal computed by behaviorMate and the ground-truth position tracked using computer vision. (**E**) Box plot describing the absolute difference between the current position of the mouse according to behaviorMate and the computer vision benchmark at each point in time (position error $= 4.1258 \pm 3.2558$ mm, mean $\pm$ std). (**F**) A 2D projection of the displays which will resemble what the test animal will see. When viewed on the actual monitors, the scene will appear more immersive. (**G**) A virtual reality (VR) scene as the subject moves down the virtual track. Modifying a few numerical parameters in the settings files allows one to change the appearance and view angle of a scene. Bottom: A scene with all red color shaders removed. (**H**) Left: Top view of implant. This side will make contact with microscope objective. Right: Bottom view. This side will make contact with the mouse's brain. (**I**) Sketch of how the implant will appear after being surgically attached to skull.

The online version of this article includes the following figure supplement(s) for figure 4:

**Figure supplement 1.** Alternative virtual reality (VR) system.

is a single-board computer equipped with an Amlogic S905X3 Quad-Core Cortex-A55 (2.016 GHz) ARMv8-A Processor, 4 GiB of DDR4 RAM, Mali-G31 MP2 GPU, and 1 LAN port, all mounted on a single PCB and running the Android operating system. Each display–ODROID pair runs an instance of the VRMate program, which means scene rendering is done independently by dedicated hardware for each monitor. This allows for scalability to any number of additional displays without compromising frame rate or resolution. Prior to starting an experiment, behaviorMate supplies each instance of VRMate with all the information it needs to correctly display the virtual environment from a particular

point of view (*Figure 4F*). Notably, Unity is platform independent, so VRMate can be built for Windows, OSX, and other systems. Therefore, it is also not strictly necessary to use dedicated Android devices, and it is possible to run behaviorMate and VRMate on the same PC. Various setups can be configured to meet individual researchers' needs.

Specifically for researchers conducting imaging experiments, Unity supports the implementation of custom 'shader programs' which can be enabled through the UI. Implementing custom shaders allows you to edit color-pixel values in the visual scenes just prior to when they are rendered. For example, this means it is possible to remove a particular color emission from the scenes or display with just a single color (*Figure 4G*). This is particularly useful for experiments where green or red fluorescent proteins are used to indicate neuronal activity. In this case, rendering scenes with the green or red component of each pixel set to 0 minimizes the risk of artifacts in the recorded data.

Instead of having separate applications for scene rendering and task logic using VRMate and behaviorMate, respectively, some systems may involve a single application that implements task logic, scene rendering, and communication with electronic controllers such as an Arduino. This approach may be appropriate in certain use cases, however, the recommended behaviorMate architecture has several important advantages. First, by rendering each viewing angle of a scene on a dedicated device, performance is improved by splitting the computational costs across several inexpensive devices rather than requiring specialized or expensive graphics cards in order to run. Moreover, the overall system becomes more modular and easier to test and debug especially when designing multiple different behavioral paradigms. behaviorMate tells VRMate what scene to display, the viewing angle, and the current position while VRMate handles the rest – in this way separating the task controller logic from the VR display modules. Third, implementing task logic in Unity would require understanding Object-Oriented Programming and C# (or learning some other 3D development platform) which is not always accessible to researchers that are typically more familiar with scripting in Python and Matlab. behaviorMate does not necessitate programming because it uses configuration files that can be modified in a standard text editor. Finally, some experiments may not require visual cues at all, in which case a system where the VR component is optional and not tightly integrated with the application is desirable. Of course, in the current implementation VRMate could still be configured to run on a single PC with the display stretched across multiple monitors. This would mitigate any concerns about potential frame rate offsets between screens and simplify the network setup. However, our testing shows the display to be smooth and continuous as VRMate is a lightweight application and the scenes are not graphically intensive and runs at a high frame rate on the suggested hardware.

## Frame and display setup

Our standard setup for immersive mouse VR experiments contains five monitors surrounding the animal. We supply plans for a VR frame to construct this (*Figure 4A*, *also see* Appendix 1). The VR frame is designed with extruded aluminum rails and holds the displays at the proper angle and distance from the animal. The goal is to completely encompass the animal's visual field with the virtual scene, so the displays are placed in an octagonal setup with the animal at the center. The angle between each display is 135 degrees and the distance between the mouse and the center display is 15 inches. Since mice primarily have a visual field-oriented overhead (*Dräger and Olsen, 1980*), the mouse is positioned 5 inches above the bottom edge of each monitor in order to align the visual stimulus to the animal's field of view.

In cases when the standard 5 monitor setup will not fit due to space constraints, we use a wireless, tablet-based setup (*Figure 4—figure supplement 1A*). When VRMate is built for Android OS, it can be installed on any Android tablet. Our tablet system consists of 5 'Fire HD 10' (2019, 9th Gen), 10.1 inch tablets with a screen resolution of 1920 × 1200 IPS while running Android 9. Additional specs include 2 GB RAM, 4xARM Cortex-A73 (2.0 GHZ) and 4xARM Cortex-A53 (2.0 GHz) processors, and an ARM Mali-G72 MP3 GPU. Tablets may be connected to the behaviorMate intranet using a wireless router. The PC running behaviorMate connects through this router to interface with the displays. Each device on the network is assigned a unique static IP address by configuring the IP tables on the Wi-Fi-enabled router to bind each tablet's IP address to its MAC address. A low-latency router with multiple antennae is recommended and should be placed as close as possible to the tablet displays.

Typical VR setups position animals on top of a running wheel which is less bulky and mechanically simpler than the fabric belt TM system. The running wheel design used in our setup, as described

previously by *Warren et al., 2021* is lightweight and optimized to have low rotational momentum. This provides animals with a natural feeling of running while minimizing the effort required to traverse the virtual environments. Mice are head fixed on top of the wheel using custom head restraining bars and a matching holder using a dove-tail and clamping system. The implant holders (*Figure 4—figure supplement 1B*) are two L-shaped components machined from solid steel. They have two M6 holes for being attached to pillar posts of aluminum rails. In addition, the implant holders each have a channel which allows the clamps to slide and accommodate head-bars of different lengths. Both the clamps and clamp holders have thumb screws not shown in the figure. The screw for the clamps tightly grips the head-bar and thus minimizes motion artifacts in imaging data. The screw for the clamp holders fixes the clamps' position and reduces motion artifacts. The head restraining bars for mice presented with visual cues (*Figure 4H, I*) are designed to assist with blocking light that may otherwise be detected during imaging. These can be 3D printed from titanium and reused between animals if properly cleaned and sterilized.

In some cases, a modified running wheel design with goniometers is required to tilt both the implant holders and the entire wheel along the roll and pitch axes (*Figure 4B*). For head-fixed 2p imaging, this is used to align the imaging plane to maximize the size of the field of view or, as needed, level the glass coverslip placed over the brain (which minimizes the amount of the material the beam must travel through and maximizes image clarity). For this modified design, aluminum rails are arranged in a 'U' shape and then placed on top of two goniometers. The implant holders sit on top of these rails, and the wheel is suspended below the rails, offset from the goniometers. The entire assembly sits on sliding mounts connected to pillar posts so the elevation of the animal's head can be adjusted.

## Example setup

A simple VR system could have two displays in an arrow configuration directly in front of a mouse which is head fixed and mounted on top of a running wheel. Attached to the back of each monitor would be a small Android computer (ODROID) connected via an HDMI cable. The left screen could be given an IP address of 192.168.1.141. The right machine will be assigned 192.168.1.142. Static IP assignment is done through the Android devices' settings menu. Arduinos running the Behavior and Position Controllers are given the IPs of 192.168.1.101 and 192.168.1.102, respectively, which is the default in our supplied code (*see* Appendix 1). Finally, a PC running the UI is assigned an IP of 192.168.1.100. We use standard IP address and ports for each of the displays and electronics; however, the IPs that the behaviorMate UI will use are specified in the settings file and determined at run-time (the Arduino IP assignments are made when the devices are loaded). Each device is also assigned a port which can be selected arbitrarily and can be changed as needed. The only requirement is that all devices are on the same subnet. In this example setup, all devices share the same network prefix of 192.168.1.x so they are part of the same subnet and will be able to communicate. Given that all components are connected to the same local network, usually via an unmanaged switch, packets will arrive to their intended recipient.

When the animal runs, turning the wheel and rotary encoder, the behaviorMate UI will receive and integrate movement updates from the Position Controller. Once the experiment is started from the UI, the PC will evaluate which contexts are active and send position updates to both screens simultaneously. The VRMate instances will receive these packets and render their view of the same scene independently. Latency between the UI and the computers running VRMate is fairly low (*Figure 4C*) so this does not present an issue. As the animal runs, it enters a reward zone. behaviorMate will send a message to the Behavior Controller triggering the 'reward context', causing a reward valve to release a drop of water or sucrose reward. As the mouse runs, the visual scene updates accordingly. Once the mouse travels a linear distance equal to the 'track_length' parameter specified in the settings file, behaviorMate advances the lap count and resets the position to 0. Between-lap 'time outs' or dark periods may also be configured. Furthermore, the user may choose to run a 'context-switch' experiment: this type of experiment might involve one scene being displayed on odd laps and another on even laps. If a scene change is necessary, between laps, behaviorMate will send a message to both running instances of VRMate that specify the switch to the alternate scene.

## VR system performance

One key concern with closed-loop control VR experimental setups is the latency between when the animal moves the physical wheel and when the displays update position in the virtual environment. We therefore performed a benchmarking test on our standard 5 monitors VR/running wheel setup (with scenes rendered by ODROIDs connected to the UI via an unmanaged Ethernet switch). An impulse was applied to the running wheel, and the time taken for the screens to update was observed. The screen delay was measured using a high-speed 150 fps camera. Latency was sampled 20 times and found to have a median value of 0.067 s, with no sample exceeding 0.087 s (*Figure 4C*). Since mouse reaction times are approximately 0.1 s (*Dombeck and Reiser, 2012*; *Mauk and Buonomano, 2004*), VR position updates using this system are nearly instantaneous to the mouse. Moreover, because position updates are constantly being sent by behaviorMate to VRMate and VRMate is immediately updating the scene according to this position, the most the scene can become out of sync with the mouse's position is proportional to the maximum latency multiplied by the running speed of the mouse. Such a degree of asynchrony is almost always negligible.

As with the fabric TM, another characteristic that should be considered for the running wheel is the accuracy of the position tracking system. Accuracy refers to how closely the mouse's virtual position recorded by behaviorMate matches the true linear distance it has run on the wheel. To find the 'true' linear distance traveled, a small dim LED was attached to the side of the running wheel, and the whole setup was placed in a dark enclosure, so the only significant source of light was the LED. The wheel was advanced by hand, and using OpenCV (*Bradski, 2000*), the total distance traversed by the LED was extracted. Concurrently, behaviorMate was detecting the current position using the attached rotary encoder. Graphs of position versus time from the data collected using computer vision and the data recorded by behaviorMate were overlaid. From *Figure 4D*, it is clear that there is little discrepancy between the two positions. Across all time points, the mean difference between the true and recorded position was 4.1 mm.

## Use in past experiments

While the introduction of a purely visual VR option for behaviorMate represents one of the newest updates to the setup, studies have already been published that specifically leverage this capability, primarily to examine the effect of rapid within-session context changes (*Priestley et al., 2022*; *Bowler and Losonczy, 2023*). VR experiments have the ability to transport animals to completely novel scenes which has been useful for investigating the mechanisms of synaptic plasticity, since it is possible to capture the animals first exposure to an environment in a controlled way (*Priestley et al., 2022*). Since the visual VR setup is integrated into behaviorMate, it is also possible to test how animals respond to changes in the rules governing reward delivery and visual scene presentation. The system further allows for simultaneous delivery of multi-modal stimuli simultaneously such as audio and visual. Thus, our system comprises an adaptable framework for probing the function of navigational circuits in the brain and the relationship between goal and context representations (*Bowler and Losonczy, 2023*).

## Experimental validation

To investigate if spatially tuned 'place cells' (*O'Keefe and Dostrovsky, 1971*) with similar properties emerge in both the belt TM system and the VR running wheel system, we performed 2p imaging in CA1 of the hippocampus while animals explored in either VR or TM. Mice were injected with GCaMP8m (AAV1-syn-GCaMP8m-WPRE, Addgene) and implanted with a glass window to have optical access to CA1 (*Figure 5*). Once mice recovered from surgeries, they were trained to do a spatial navigation task in either VR or TM. After the mice were fully trained, 2p imaging was performed during navigation. Calcium signals were processed with Suite2p for motion correction, cell segmentation, and fluorescent signal and neuropil extraction (*Pachitariu et al., 2017*). Extracted raw fluorescent signals were corrected for neuropil contamination, and resulting $\Delta F/F$ was used to estimate spike events using OASIS (*Friedrich et al., 2017*). A median filter was applied to binarize spike amplitudes into bins of position, and all subsequent analyses used this binarized train as an estimation of each ROIs activity. We recorded on average 1533 ± 481 and 1057 ± 169 CA1 neurons in VR and TM, respectively. Although VR had significantly fewer CA1 neurons being classified as place cells compared to TM (VR: 34.9% (3253/9195), TM: 44.9% (6101/13,454), two sample $z$-test, $z$ = 14.96, p = 1.27e−50), place cells

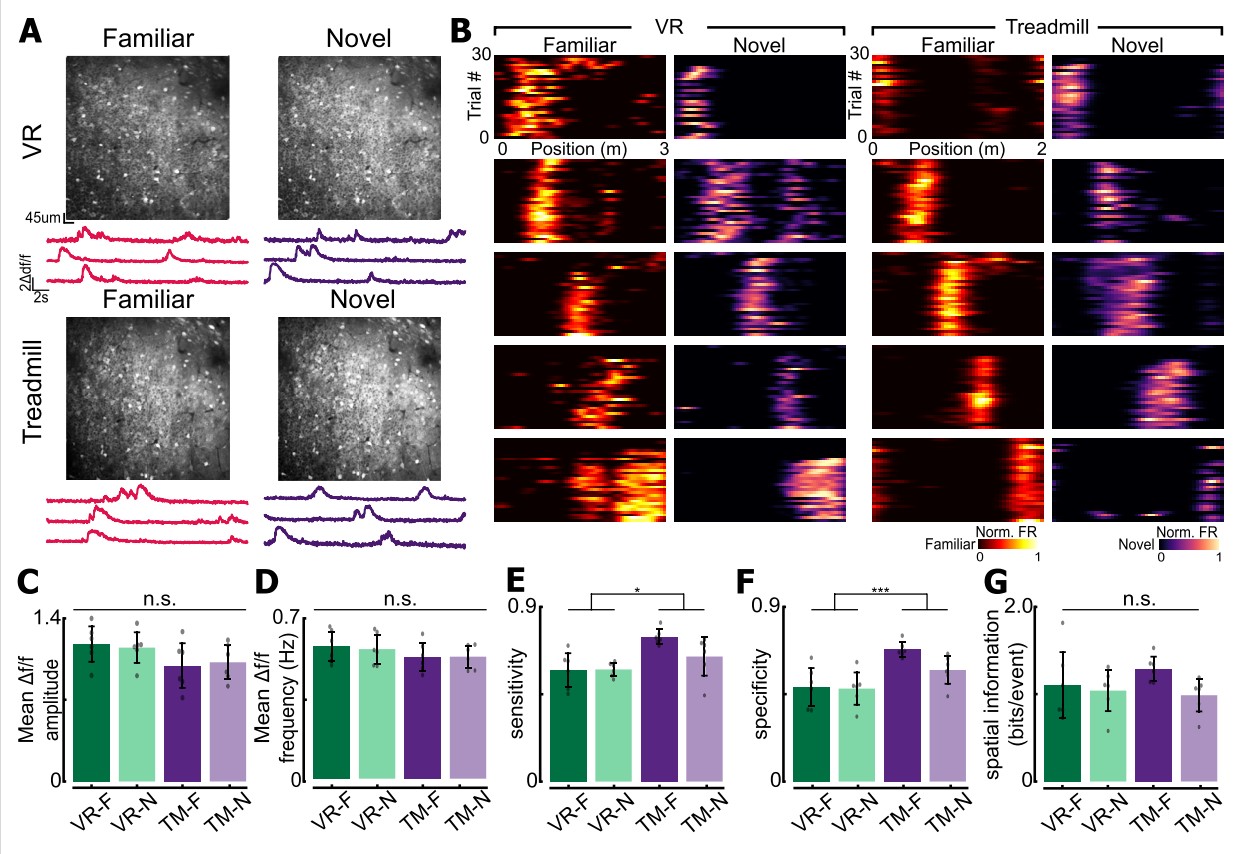

**Figure 5.** Two-photon imaging of CA1 population for experimental validation. (**A**) Field of view obtained from virtual reality (VR) and treadmill (TM). (**B**) Examples of place cells recorded using VR or TM. In both familiar and novel contexts, place cells encoding different locations on the track are found. Note that they are different place cells from individual animals. (**C**) Mean Δf/f amplitude of place cells recorded each context–environment pair (F = familiar context, N = novel context). (**D**) Mean Δf/f frequency of place cells. For (**C, D**), only significant Δf/f events were used. See methods for details. (**E**) Place sensitivity in each context–environment pair. Significant main effect of environment (VR vs. TM, p = 0.0257). (**F**) Place cell specificity in each context–environment pair. Significant main effect of environment (VR vs. TM, p = 0.008). (**G**) Spatial information (bits/event) in each context–environment pair. Two-way repeated measures of ANOVA were used unless otherwise stated. Refer to Table *Figure 5—figure supplement 2* for ANOVA table. For panels C–G, n = 6 mice (all mice were recorded in both VR and TM systems), 3253 cells in VR classified as significantly tuned place cells VR, and 6101 tuned cells in TM. For all panels, n.s. = non-significant, *p < 0.05, ***p < 0.001.

The online version of this article includes the following source data and figure supplement(s) for figure 5:

**Figure supplement 1.** Additional place cell properties.

**Figure supplement 2.** ANOVA results table of ANOVA results relating to *Figure 5*, *Figure 5—figure supplement 1*.

**Figure supplement 2—source data 1.** Data file containing a digital copy of the information presented in *Figure 5—figure supplement 2*.

tiled the entire track in both VR and TM (*Figure 5B*, *Figure 5—figure supplement 1*). In all subsequent analyses, we used CA1 neurons identified as a place cell unless otherwise stated.

To assess differences in the calcium signal of place cells found in context–environment pairs, we first compared the Δ*F/F* amplitude and frequency of individual ROIs identified as place cells. Only Δ*F/F* events that were 3 standard deviations above the mean Δ*F/F* were included in these analyses. To examine the effect of environment (VR vs. TM) and context (familiar (F) vs. novel (N)) on Δ*F/F* amplitude and frequency, we conducted two-way repeated measures of ANOVA with both environment and context as a within-subject factors. There was no significant effect of environment or context on mean Δ*F/F* amplitude (environment: $F_{(1,5)} = 1.9387$, p = 0.222, context: $F_{(1,5)} = 0.0037$, p = 0.9537, *Figure 5C*, *Figure 5—figure supplement 2*) or on mean Δ*F/F* frequency (environment: $F_{(1,5)} = 4.37$, p = 0.091, context: $F_{(1,5)} = 0.0789$, p = 0.7901, *Figure 5D*, *Figure 5—figure supplement 2*). These results suggest that calcium transient properties of place cells observed in both environments and contexts were similar. Next, we quantified individual place cells' specificity, sensitivity, and spatial

information to identify any differences in place cell firing properties between context–environment pairs. As described above, two-way repeated measures of ANOVA were used with both context and environment as within-subject factors. The two-way ANOVA revealed a significant effect of environment on sensitivity (environment: $F(1,5) = 9.85$, p = 0.0257, context: $F(1,5) = 3.34$, p = 0.13, *Figure 5E*, *Figure 5—figure supplement 2*) and specificity (environment: $F(1,5) = 53.32$, p = 0.0008, context: $F(1,5) = 2.55$, p = 0.17, *Figure 5F*, *Figure 5—figure supplement 2*), suggesting that place cells in TM have higher trial-by-trial in-field firing compared to those from VR. However, we did not observe any significant effect of environment or context on spatial information (*Skaggs and McNaughton, 1998*) (environment: $F(1,5) = 0.74$, p = 0.43, context: $F(1,5) = 3.79$, p = 0.11, *Figure 5G*, *Figure 5—figure supplement 2*). These results are consistent with previous findings that place cells in VR have reduced trial-by-trial reliability (*Ravassard et al., 2013*; *Aghajan et al., 2015*), but still encode spatial information (*Dombeck et al., 2010*; *Aronov and Tank, 2014*; *Hainmueller and Bartos, 2018*; *Priestley et al., 2022*). The observed differences in the properties of VR and TM place cells may be the result of the absence of proximal visual cues and tactile cues (which are speculated to be more salient than

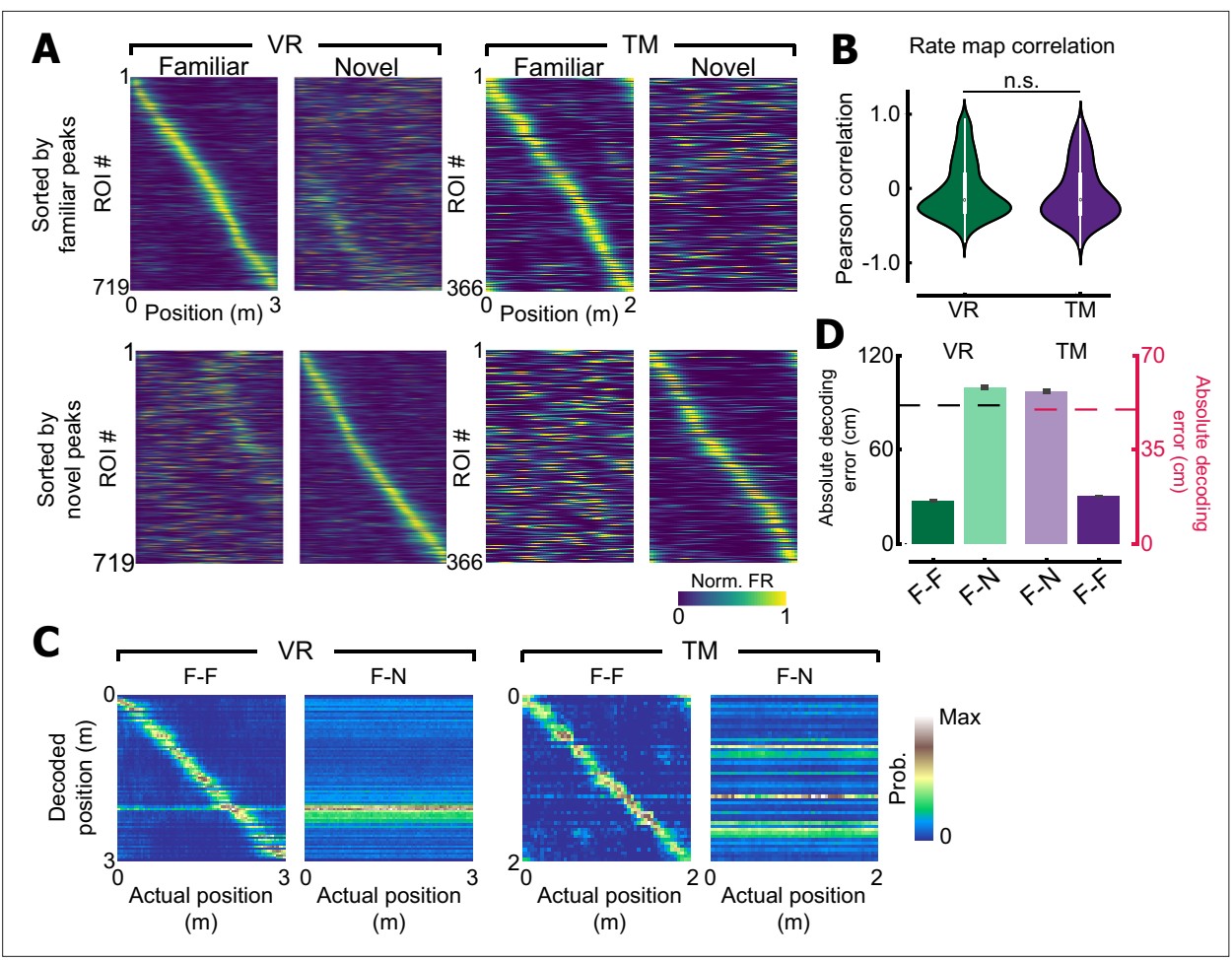

**Figure 6.** Place cells remap in both environments. (**A**) Place cells recorded in virtual reality (VR) or treadmill (TM) sorted by their familiar peak (top) or novel peak (bottom) locations. A subset of CA1 neurons remaps when exposed to novel context in both VR and TM. (**B**) Rate map correlation of individual place cells between familiar and novel contexts. No difference was observed between VR and TM (Wilcoxon rank-sum test, F = 1.38, p = 0.17). (**C**) Confusion matrices of the Bayesian decoder for VR and TM. F–F indicates a model trained and tested with trials from the familiar context. F–N is a model trained with trials from familiar context, and tested with a novel trial. (**D**) Absolute decoding error for the maximum likelihood estimator. A dashed line indicates a chance level (88 ± 1.96 cm for VR/ 50 cm for TM, see methods). Note that when the decoder is trained and tested with familiar context trials, absolute decoding accuracy is below chance in both VR and TM (VR: 27.62 ± 1.44 cm, p < 0.01, TM: 17.78 ± 0.44 cm, p < 0.01, permutation test). However, when the decoder is trained with familiar context trials, but tested with novel context trials, decoding accuracy was increased to the chance level (VR: 99.80 ± 3.95 cm, p = 1.00, TM: 56.82 ± 2.46 cm, p = 1.00).

visual cues) during VR tasks (*Knierim and Rao, 2003*; *Renaudineau et al., 2007*; *Sofroniew et al., 2014*).

Next, we wanted to investigate if place cells remap in both VR and TM environments when mice are exposed to a novel context (*Muller and Kubie, 1987*; *Leutgeb et al., 2005*). As expected, a subset of place cells changed their place tuning properties when exposed to novel contexts (*Figure 6A, B*). Moreover, we did not observe differences in the between context rate map correlation between VR and TM, suggesting that remapping happens in both environment types. To further quantify the extent of remapping in each environment, we built a naive Bayes decoder to decode position based on population activity of CA1 neurons. When the decoder was trained with population activity from familiar trials and tested with population activity from a held-out familiar trial, the absolute decoding error was significantly below chance ($p < 0.05$) in both VR and TM. However, when the decoder trained with familiar trials was tested to decode position in a novel trial, the absolute decoding error was at the chance level (*Figure 6C, D*) in both VR and TM. These suggest that place cells remap in both VR and TM when exposed to novel context.

## Discussion

behaviorMate is, most generally, a solution for controlling and collecting feedback from devices in behavioral experiments. The behaviorMate methodology is built around an 'Intranet of Things' approach, expanding to incorporate additional components to meet experimental demands but also compatible with sparser setups to ease complexity when not needed. It has typically been used concurrently with neuronal imaging, since the timestamped output of behavior events is relatively simple to align with imaging data, providing experimenters with the ability to correlate neuronal events with behavior. In addition, behaviorMate is designed to be flexible, expanding to incorporate additional components as specified in the settings while making minimal assumptions. Experimenters may incorporate other techniques such as imaging and electrophysiology recordings or just run behavior. For experimental paradigms involving any combination of displays, 1D movement (wheels, TMs, etc.), sensors (LEDs, Capacitance lick sensors, etc.), and actuators (odor release systems, solenoid valves, etc.), this is a cost-effective and reliable solution.

The main modules of behaviorMate are the Java GUI application, referred to as the UI, Behavior Controller, and Position Controller. The UI coordinates most communication between devices, although sometimes devices directly communicate with each other for performance reasons. The Behavior Controller sends messages to and receives data from nearly any kind of peripheral devices such as 2p microscopes, LEDs, speakers, computer displays, and electrophysiology systems. The Position Controller is solely responsible for receiving position updates from the rotary encoder. Any number of additional devices may be used by connecting them to the behavior controller using standard connectors. Furthermore, the companion software of VRMate allows for low-cost integration of VR experiments that are already compatible with behaviorMate. The appeal of behaviorMate is its modularity, reliability, and open-source code base.

Finally, many published studies have relied on this system and the resulting data collected (*Bowler and Losonczy, 2023*; *Tuncdemir et al., 2023*; *Vancura et al., 2023*; *Priestley et al., 2022*; *O'Hare et al., 2022*; *Rolotti et al., 2022a*; *Tuncdemir et al., 2022*; *Rolotti et al., 2022b*; *Terada et al., 2022*; *Geiller et al., 2022*; *Blockus et al., 2021*; *Grosmark et al., 2021*; *Rolotti et al., 2022a*; *Geiller et al., 2020*; *Kaufman et al., 2020*; *Turi et al., 2019*; *Zaremba et al., 2017*; *Danielson et al., 2017*; *Danielson et al., 2016*). These paradigms varied greatly and included random foraging, goal-oriented learning, multi-modal cue presentations, and context switches (both in VR environments and physical TMs). Instructions for downloading the UI software and the companion VRMate software, and assembling a Behavior Controller, Position Controller, running wheel, VR monitor frame, and TM, can be found on the Losonczy Lab website (https://www.losonczylab.org/software).

# Materials and methods

## Lead contact and materials availability

Further information and requests for resources and reagents should be directed to the Lead Contact Attila Losonczy (al2856@columbia.edu). All unique resources generated in this study are available from the Lead Contact with a completed Materials Transfer Agreement.

## Experimental model and subject details

### Animals

All experiments were conducted in accordance with the NIH guidelines and with the approval of the Columbia University Institutional Animal Care and Use Committee (Protocol #AC-AABF6554). Experiments were performed with six adult mice between 8 and 16 weeks. Five mice were Unc5b crossed with C57Bl/6, and one mouse was VGAT-cre crossed with Ai9 (Jackson Laboratory). Three female and three male mice were used in this experiment to balance for sex.

### Surgical procedures

Surgeries were done as previously described (*Lovett-Barron et al., 2014*). Briefly, animals were anesthetized with isoflurane and injected with jGCaMP8m (rAAV1-syn-GCaMP8m-WPRE, Addgene: 162375-AAV1) in dorsal CA1 (AP: 2.1, ML: 1.4, DV: 1.3, 1,2, 1,1, 100 µl per site) using Nanoject III (Drummond). Animals were allowed to recover from the injection surgery for 7 days, and the cortex lying above CA1 was aspirated to implant a glass cannula to enable optical access to CA1. Finally, a titanium headpost was cemented to the skull using dental cement to head-fix animals for 2p functional imaging. Analgesics were provided as per Columbia University Institutional Animal Care and Use Committee prior and after the surgeries for up to 3 days.

### 2p imaging and behavior

After recovering from surgeries, animals were habituated to head fixation in a VR environment for 15 min. Subsequently, animals were trained to run on a lightweight wheel, and 5% sucrose was given at random locations to motivate animals' to run in the VR or TM. When animals reliably ran at least 60 laps in VR, or 30 laps in TM, the reward was fixed to either one or two locations. A 3-m VR track was used for all animals except jy030, which had 4-m VR track instead, and 2-m TM belt was used in all animals. For VR experiments, animals were imaged for one session, and the context switch happened after animals ran 30 laps in familiar context. Animals were required to run at least another 30 laps in the novel context. In TM, imaging was done over 2 days (first day: familiar belt, second day: novel belt). 2p imaging was performed using a 8-kHz resonant scanner (Bruker) and a 16× Nikon water immersion objective (0.8 NA, 3 mm working distance). In order to image GCaMP8m signals, 920 nm excitation wavelength was used (Coherent), and the power from the tip of the objective never exceeded 100 mW. GCaMP signals were collected using a GaAsP photomultiplier tube detector (Hamamatsu, 7422P-40) following amplification via a custom dual stage preamplifier (1.4 × 105 dB, Bruker). All imaging was performed using 512 × 512 pixels with digital zoom between 1 and 1.4.

## Calcium signal processing

Imaging data was processed as previously described (*Priestley et al., 2022*). In short, the SIMA software package (*Kaifosh et al., 2014*) was used to organize the imaging dataset, and the imaging data was processed with Suite2p (*Pachitariu et al., 2017*) for motion correction, signal/neuropil extraction, and ROI detection. Following ROI detection, individual ROIs were manually curated using the Suite2p GUI to exclude non-somatic ROIs. $\Delta F/F$ signals were calculated after subtracting the neuropil and correcting for baseline.

## Data analysis

As previously described, all analyses were done using binarized event signals (*Ahmed et al., 2020*; *Priestley et al., 2022*). Briefly, $\Delta F/F$ signals were deconvolved using OASIS (*Friedrich et al., 2017*), and the resulting estimated spike amplitude train was binarized by selecting events whose amplitudes were above 4 median absolute deviations of the raw trace. We do not claim that this reflects the true spiking activities of individual ROIs. Binarized event trains were binned into 4 cm bins and smoothed

with a Gaussian filter (SD = 2) to calculate tuning curves. Individual ROIs were classified as a place cell by identifying bins that have activities greater than the 99th percentile of surrogate tuning curves (*Priestley et al., 2022*). In short, surrogate average tuning curves were calculated by performing 1000 circular shuffles of their occupancy of individual trials. Bins that had activity greater than 99th percentile of the tuning curves were identified as potentially significant bins. In order to be classified as a place cell, ROIs needed to have at least 3 consecutive significant bins (12 cm), but less than 25 consecutive bins (100 cm). Moreover, in order to avoid spurious detection of significant bins, binarized events must have been detected in at least 50% of trials within those significant bins.

For mean $\Delta F/F$ amplitude and frequency calculations, only bins with 3 standard deviations above the mean were used. For place cell sensitivity, representing the proportion of laps that had active events within the significant field, the number of laps that had at least one binarized event within the significantly detected fields was divided by the total number of laps. For place cell specificity, total number of events within the significant field was divided by the total number of events observed within the lap. Then, it was averaged across all laps to have a single value for each ROI. If multiple fields were detected, they were computed separately and averaged across fields to have a single value for each ROI. For spatial information, it was calculated as described previously (*Skaggs and McNaughton, 1998*). For remapping analysis, one animal was used for the analysis (jy065). Given that TM imaging happened across 2 days, FOVs recorded from TM were matched using CellReg (*Sheintuch et al., 2017*), and spatial mask correlation was used to find the same ROIs. Results were manually curated within the Suite2p GUI. This was not necessary for VR since the context switch happened within the same session. To calculate rate map correlation, the Pearson correlation coefficient was computed between average tuning curves from the familiar and novel contexts. For TM, given that place cells may anchor on seams of the belt, the shift that generated the maximum population vector correlation between familiar and novel contexts was calculated. All ROIs recorded in the novel context were circularly shifted by the same amount, and the rate map correlation was calculated as described above. A naive Bayesian classifier was used to decode animals' position on the track (*Zhang et al., 1998*). For each frame and spatial bin:

$$P(pos|a_{all}) = C \left( \prod_{i=1}^{N} f_i(pos)^{a_i} \right) e^{-\tau \sum_{i=1}^{N} f_i(pos)}$$

where $N$ is the total number of cells, $f_i(pos)$ is the average binned activity of a cell, $\tau$ is the bin size, and $a_i$ is the activity on the frame. $C$ is the normalization constant. The position bin with the highest probability was selected as a decoded position. Chance level for VR was calculated by randomly shuffling cell labels of the testing set. For TM, a chance level was determined as track length/4, which is a chance level of a circular environment (50 cm). The decoder was trained with $n - 1$ trials and tested on the left out trial. To match ROI numbers recorded between VR and TM, 500 ROIs were randomly selected, and the decoding accuracy was calculated 50 times. Average absolute decoding accuracy was compared to the chance level calculated from each iteration, and the number of times that it was below the chance level was considered to be its p value.

## Acknowledgements

We thank Drs. Ali Kaufman and Ally Lowell for troubleshooting and debugging the initial iterations of behaviorMate as well as Dr. Andres Grosmark for continued feedback on the project. Additionally, we thank Dr. Stephanie Herrlinger, Abhishek Shah, and all other members of the Losonczy lab for comments and discussion on the manuscript.

## Additional information

### Funding

| Funder | Grant reference number | Author |
| --- | --- | --- |
| National Institute of Mental Health | NIMH F31NS110316 | John C Bowler |

| Funder | Grant reference number | Author |
|---|---|---|
| National Institute of Mental Health | R01MH124047 | Attila Losonczy |
| National Institute of Mental Health | R01MH124867 | Attila Losonczy |
| National Institute of Neurological Disorders and Stroke | R01NS121106 | Attila Losonczy |
| National Institute of Neurological Disorders and Stroke | U01NS115530 | Attila Losonczy |
| National Institute of Neurological Disorders and Stroke | R01NS133381 | Attila Losonczy |
| National Institute of Neurological Disorders and Stroke | R01NS131728 | Attila Losonczy |
| National Institute on Aging | RF1AG080818 | Attila Losonczy |
| EPFL ELISIR | | James B Priestley |
| Fondation Marina Cuennet-Mauvernay | | James B Priestley |

The funders had no role in study design, data collection and interpretation, or the decision to submit the work for publication.

### Author contributions

John C Bowler, Conceptualization, Data curation, Software, Formal analysis, Investigation, Methodology, Writing – original draft, Writing – review and editing; George Zakka, Data curation, Software, Formal analysis, Investigation, Methodology, Writing – original draft, Writing – review and editing; Hyun Choong Yong, Data curation, Software, Formal analysis, Investigation, Methodology, Writing – original draft; Wenke Li, Conceptualization, Software, Investigation, Methodology; Bovey Rao, Investigation, Methodology, Writing – review and editing; Zhenrui Liao, Software, Investigation, Methodology, Writing – review and editing; James B Priestley, Data curation, Software, Investigation, Methodology, Writing – original draft; Attila Losonczy, Resources, Supervision, Methodology, Writing – original draft, Project administration, Writing – review and editing

### Author ORCIDs

John C Bowler ⓘ https://orcid.org/0000-0003-0498-5743
George Zakka ⓘ https://orcid.org/0009-0005-0035-6749
Hyun Choong Yong ⓘ https://orcid.org/0000-0002-1841-6317
Attila Losonczy ⓘ https://orcid.org/0000-0002-7064-0252

### Ethics

This study was performed in strict accordance with the recommendations in the Guide for the Care and Use of Laboratory Animals of the National Institutes of Health. All of the animals were handled according to approved Institutional Animal Care and Use Committee (IACUC) protocols of Columbia University (#AC-AABF6554). The protocol was approved by the Committee on the Ethics of Animal Experiments of Columbia University. All surgery was performed under sodium isoflurane anesthesia, and every effort was made to minimize suffering.

Reviewer #1 (Public review): https://doi.org/10.7554/eLife.97433.3.sa1
Reviewer #2 (Public review): https://doi.org/10.7554/eLife.97433.3.sa2
Reviewer #3 (Public review): https://doi.org/10.7554/eLife.97433.3.sa3
Author response https://doi.org/10.7554/eLife.97433.3.sa4

## Additional files

**Supplementary files**
MDAR checklist

### Data availability

Requests for further information may be directed to the Lead Contact Attila Losonczy (al2856@columbia.edu). Data referenced in Figures 3–6 are openly hosted on Dryad at https://doi.org/10.5061/dryad.02v6wwqdf. See Appendix 1 for links to all software referenced by this manuscript.

The following dataset was generated:

| Author(s) | Year | Dataset title | Dataset URL | Database and Identifier |
| --- | --- | --- | --- | --- |
| Bowler J, Zakka G, Yong H, Li W, Rao B, Liao Z, Priestley JB, Losonczy A | 2025 | behaviorMate: An Intranet of Things Approach for Adaptable Control of Behavioral and Navigation-Based Experiments | https://doi.org/10.5061/dryad.02v6wwqdf | Dryad Digital Repository, 10.5061/dryad.02v6wwqdf |

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

## Appendix 1

### Key links

- Link to website for quickstart guides and file downloads:https://www.losonczylab.org/behaviormate
- Link to example settings files:https://github.com/losonczylab/behaviorMate/tree/main/example_settings_files
- Link to source code for the UI:https://github.com/losonczylab/behaviorMate
- Link to online java documentation for the UI:https://www.losonczylab.org/behaviorMate-1.0.0
- Link to electronics schematics:https://github.com/losonczylab/Hardware/tree/main/Electronics
- Link to Arduino Firmware and Debugging Utilities:https://github.com/losonczylab/behaviormate_utils/
- Link to treadmill and VR CAD files: https://github.com/losonczylab/Hardware/
- Link to benchmarking and experimental validation data:https://doi.org/10.5061/dryad.02v6wwqdf

