## [Editor Report · eLife Assessment]

Bowler et al. present a software/hardware system for behavioral control of navigation-based virtual reality experiments, particularly suited for pairing with 2-photon imaging but applicable to a variety of techniques. This system represents a **valuable** contribution to the field of behavioral and systems neuroscience, as it provides a standardized, easy to implement, and flexible system that could be adopted across multiple laboratories. The authors provide **compelling** evidence of the functionality of their system by reporting benchmark tests and demonstrating hippocampal activity patterns consistent with standards in the field. This work will be of interest to systems neuroscientists looking to integrate flexible head-fixed behavioral control with neural data acquisition.

---

## [Referee Report · Reviewer #1 (Public review)]

Summary:

Bowler et al. present a thoroughly tested system for modularized behavioral control of navigation-based experiments, particularly suited for pairing with 2-photon imaging but applicable to a variety of techniques. This system, which they name behaviorMate, represents an important methodological contribution to the field of behavioral and systems neuroscience. As the authors note, behavioral control paradigms vary widely across laboratories in terms of hardware and software utilized and often require specialized technical knowledge to make changes to these systems. Having a standardized, easy to implement, and flexible system that can be used by many groups is therefore highly desirable.

Strengths:

The present manuscript provides compelling evidence of the functionality and applicability of behaviorMate. The authors report benchmark tests for high-fidelity, real-time update speed between the animal's movement and the behavioral control, on both the treadmill-based and virtual reality (VR) setups. The VR system relies on Unity, a common game development engine, but implements all scene generation and customizability in the authors' behaviorMate and VRMate software, which circumvents the need for users to program task logic in C# in Unity. Further, the authors nicely demonstrate and quantify reliable hippocampal place cell coding in both setups, using synchronized 2-photon imaging. This place cell characterization also provides a concrete comparison between the place cell properties observed in treadmill-based navigation vs. visual VR in a single study, which itself is a valuable contribution to the field.

Weaknesses: None noted.

Documentation for installing and operating behaviorMate is available via the authors' lab website and Github, linked in the manuscript.

The authors have addressed all of my requests for clarification from the previous round of review. This work will be of great interest to systems neuroscientists looking to integrate flexible head-fixed behavioral control with neural data acquisition.

---

## [Referee Report · Reviewer #2 (Public review)]

The authors present behaviorMate, an open-source behavior control system including a central GUI and compatible treadmill and display components. Notably, the system utilize the "Intranet of things" scheme and the components communicate through local network, making the system modular, which in turn allows user to configure the setup to suit their experimental needs. Overall, behaviorMate is a useful resource for researchers performing head-fixed VR imaging studies involving 1D navigation tasks, as the commercial alternatives are often expensive and inflexible to modify.

One major utility of behaviorMate is an open-source alternative to commercial behavior apparatus for head-fixed imaging studies involving 1D navigation tasks. The documentation, BOM, CAD files, circuit design, source and compiled software, along with the manuscript, create an invaluable resource for neuroscience researcher looking to set up a budget-friendly VR and head-fixed imaging rig. Some features of behaviorMate, including the computer vision-based calibration of treadmill, and the decentralized, Android-based display devices, are very innovative approaches and can be quite useful in practical settings.

behaviorMate can also be used as a set of generic schema and communication protocols that allows the users to incorporate recording and stimulation devices during a head-fixed imaging experiment. Due to the "Intranet of things" approach taken in the design, any hardware that supports UDP communication can in theory be incorporated into the system. In terms of current capability, behaviorMate supports experimental contingencies based on animal position and time and synchronization with external recording devices using a TTL start signal. Further customization involving more complicated experimental contingencies, more accurate recording synchronization (for example with ephys recording devices), incorporation of novel behavior and high-speed neural recording hardware beyond GPIO signaling would require modification of the Java source and custom hardware implementation. Modification to the Java source of behaviorMate can be performed with basic familiarity with object-oriented programming using the Java programming language, and a JavaFX-based plugin system is under development to make such customizations more approachable for users.

In summary, the manuscript presents a well-developed and useful open-source behavior control system for head-fixed VR imaging experiments with innovative features.

---

## [Referee Report · Reviewer #3 (Public review)]

In this work, the authors present an open-source system called behaviourMate for acquiring data related to animal behavior. The temporal alignment of recorded parameters across various devices is highlighted as crucial to avoid delays caused by electronics dependencies. This system not only addresses this issue but also offers an adaptable solution for VR setups. Given the significance of well-designed open-source platforms, this paper holds importance.

Advantages of behaviorMate:

The cost-effectiveness of the system provided.

The reliability of PCBs compared to custom-made systems.

Open-source nature for easy setup.

Plug & Play feature requiring no coding experience for optimizing experiment performance (only text based Json files, 'context List' required for editing).

---

## [Author Response]

The following is the authors’ response to the previous reviews.

**Public Reviews:**

**Reviewer #1 (Public Review):**
(1) As VRMate (a component of behaviorMate) is written using Unity, what is the main advantage of using behaviorMate/VRMate compared to using Unity alone paired with Arduinos (e.g. Campbell et al. 2018), or compared to using an existing toolbox to interface with Unity (e.g. Alsbury-Nealy et al. 2022, DOI: 10.3758/s13428-021-01664-9)? For instance, one disadvantage of using Unity alone is that it requires programming in C# to code the task logic. It was not entirely clear whether VRMate circumvents this disadvantage somehow -- does it allow customization of task logic and scenery in the GUI? Does VRMate add other features and/or usability compared to Unity alone? It would be helpful if the authors could expand on this topic briefly.

We have updated the manuscript (lines 412-422) to clarify the benefits of separating the VR system as an isolated program and a UI that can be run independently. We argue that “…the recommended behaviorMate architecture has several important advantages. Firstly, by rendering each viewing angle of a scene on a dedicated device, performance is improved by splitting the computational costs across several inexpensive devices rather than requiring specialized or expensive graphics cards in order to run…, the overall system becomes more modular and easier to debug [and] implementing task logic in Unity would require understanding Object-Oriented Programming and C# … which is not always accessible to researchers that are typically more familiar with scripting in Python and Matlab.”

VRMate receives detailed configuration info from behaviorMate at runtime as to which VR objects to display and receives position updates during experiments. Any other necessary information about triggering rewards or presenting non-VR cues is still handled by the UI so no editing of Unity is necessary. Scene configuration information is in the same JSON format as the settings files for behaviorMate, additionally there are Unity Editor scripts which are provided in the VRmate repository which permit customizing scenes through a “drag and drop” interface and then writing the scene configuration files programmatically. Users interested in these features should see our github page to find example scene.vr files and download the VRMate repository (including the editor scripts). We provided 4 vr contexts, as well as a settings file that uses one of them which can be found on the behaviorMate github page (https://github.com/losonczylab/behaviorMate) in the “vr_contexts” and “example_settigs_files” directories. These examples are provided to assist VRMate users in getting set up and could provide a more detailed example of how VRMate and behaviorMate interact.

(2) The section on "context lists", lines 163-186, seemed to describe an important component of the system, but this section was challenging to follow and readers may find the terminology confusing. Perhaps this section could benefit from an accompanying figure or flow chart, if these terms are important to understand.

We maintain the use of the term context and context list in order to maintain a degree of parity with the java code. However, we have updated lines 173-175 to define the term context for the behaviorMate system: “... a context is grouping of one or more stimuli that get activated concurrently. For many experiments it is desirable to have multiple contexts that are triggered at various locations and times in order to construct distinct or novel environments.”

a. Relatedly, "context" is used to refer to both when the animal enters a particular state in the task like a reward zone ("reward context", line 447) and also to describe a set of characteristics of an environment (Figure 3G), akin to how "context" is often used in the navigation literature. To avoid confusion, one possibility would be to use "environment" instead of "context" in Figure 3G, and/or consider using a word like "state" instead of "context" when referring to the activation of different stimuli.

Thank you for the suggestion. We have updated Figure 3G to say “Environment” in order to avoid confusion.

(3) Given the authors' goal of providing a system that is easily synchronizable with neural data acquisition, especially with 2-photon imaging, I wonder if they could expand on the following features:

a. The authors mention that behaviorMate can send a TTL to trigger scanning on the 2P scope (line 202), which is a very useful feature. Can it also easily generate a TTL for each frame of the VR display and/or each sample of the animal's movement? Such TTLs can be critical for synchronizing the imaging with behavior and accounting for variability in the VR frame rate or sampling rate.

Different experimental demands require varying levels of precision in this kind of synchronization signals. For this reason, we have opted against a “one-size fits all” for synchronization with physiology data in behaviorMate. Importantly this keeps the individual rig costs low which can be useful when constructing setups specifically for use when training animals. behaviorMate will log TTL pulses sent to GPIO pins setup as sensors, and can be configured to generate TTL pulses at regular intervals. Additionally all UDP packets received by the UI are time stamped and logged. We also include the output of the arduino millis() function in all UDP packets which can be used for further investigation of clock drift between system components. Importantly, since the system is event driven there cannot be accumulating drift across running experiments between the behaviorMate UI and networked components such as the VR system.

For these reasons, we have not needed to implement a VR frame synchronization TTL for any of our experiments, however, one could extend VRMate to send "sync" packets back to behaviorMate to log when each frame was displayed precisely or TTL pulses (if using the same ODROID hardware we recommend in the standard setup for rendering scenes). This would be useful if it is important to account for slight changes in the frame rate at which the scenes are displayed. However, splitting rendering of large scenes between several devices results in fast update times and our testing and benchmarks indicate that display updates are smooth and continuous enough to appear coupled to movement updates from the behavioral apparatus and sufficient for engaging navigational circuits in the brain.

b. Is there a limit to the number of I/O ports on the system? This might be worth explicitly mentioning.

We have updated lines 219-220 in the manuscript to provide this information: Sensors and actuators can be connected to the controller using one of the 13 digital or 5 analog input/output connectors.

c. In the VR version, if each display is run by a separate Android computer, is there any risk of clock drift between displays? Or is this circumvented by centralized control of the rendering onset via the "real-time computer"?

This risk is mitigated by the real-time computer/UI sending position updates to the VR displays. The maximum amount scenes can be out of sync is limited because they will all recalibrate on every position update – which occurs multiple times per second as the animal is moving. Moreover, because position updates are constantly being sent by behaviorMate to VRMate and VRMate is immediately updating the scene according to this position, the most the scene can become out of sync with the mouse's position is proportional to the maximum latency multiplied by the running speed of the mouse. For experiments focusing on eliciting an experience of navigation, such a degree of asynchrony is almost always negligible. For other experimental demands it could be possible to incorporate more precise frame timing information but this was not necessary for our use case and likely for most other use cases. Additionally, refer to the response to comment 3a.

**Reviewer #2 (Public review):**
(1) The central controlling logic is coupled with GUI and an event loop, without a documented plugin system. It's not clear whether arbitrary code can be executed together with the GUI, hence it's not clear how much the functionality of the GUI can be easily extended without substantial change to the source code of the GUI. For example, if the user wants to perform custom real-time analysis on the behavior data (potentially for closed-loop stimulation), it's not clear how to easily incorporate the analysis into the main GUI/control program.

Without any edits to the existing source code behaviorMate is highly customizable through the settings files, which allow users to combine the existing contexts and decorators in arbitrary combinations. Therefore, users have been able to perform a wide variety of 1D navigation tasks, well beyond our anticipated use cases by generating novel settings files. The typical method for providing closed-loop stimulation would be to set up a context which is triggered by animal behavior using decorators (e.g. based on position, lap number and time) and then trigger the stimulation with a TTL pulse. Rarely, if users require a behavioral condition not currently implemented or composable out of existing decorators, it would require generating custom code in Java to extend the UI. Performing such edits requires only knowledge of basic object-oriented programming in Java and generating a single subclass of either the BasicContextList or ContextListDecorator classes. In addition, the JavaFX (under development) version of behaviorMate incorporates a plugin which doesn't require recompiling the code in order to make these changes. However, since the JavaFX software is currently under development, documentation does not yet exist. All software is open-sourced and available on github.com for users interested in generating plugins or altering the source code.

We have added the additional caveat to the manuscript in order to clarify this point (Line 197-202): “However, if the available set of decorators is not enough to implement the required task logic, some modifications to the source code may be necessary. These modifications, in most cases, would be very simple and only a basic understanding of object-oriented programming is required. A case where this might be needed would be performing novel customized real-time analysis on behavior data and activating a stimulus based on the result”

(2) The JSON messaging protocol lacks API documentation. It's not clear what the exact syntax is, supported key/value pairs, and expected response/behavior of the JSON messages. Hence, it's not clear how to develop new hardware that can communicate with the behaviorMate system.

The most common approach for adding novel hardware is to use TTL pulses (or accept an emitted TTL pulse to read sensor states). This type of hardware addition is possible through the existing GPIO without the need to interact with the software or JSON API. Users looking to take advantage of the ability to set up and configure novel behavioral paradigms without the need to write any software would be limited to adding hardware which could be triggered with and report to the UI with a TTL pulse (however fairly complex actions could be triggered this way).

For users looking to develop more customized hardware solutions that interact closely with the UI or GPIO board, additional documentation on the JSON messaging protocol has been added to the behaviormate-utils repository (https://github.com/losonczylab/behaviormate_utils). Additionally, we have added a link to this repository in the *Supplemental Materials* section (line 971) and referenced this in the manuscript (line 217) to make it easier for readers to find this information.

Furthermore, developers looking to add completely novel components to the UI can implement the interface described by Context.java in order to exchange custom messages with hardware. (described in the JavaDoc: https://www.losonczylab.org/behaviorMate-1.0.0/) These messages would be defined within the custom context and interact with the custom hardware (meaning the interested developer would make a novel addition to the messaging API). Additionally, it should be noted that without editing any software, any UDP packets sent to behaviorMate from an IP address specified in the settings will get time stamped and logged in the stored behavioral data file meaning that are a large variety of hardware implementation solutions using both standard UDP messaging and through TTL pulses that can work with behaviorMate with minimal effort. Finally, see response to R2.1 for a discussion of the JavaFX version of the behaviorMatee UI including plugin support.

(3) It seems the existing control hardware and the JSON messaging only support GPIO/TTL types of input/output, which limits the applicability of the system to more complicated sensor/controller hardware. The authors mentioned that hardware like Arduino natively supports serial protocols like I2C or SPI, but it's not clear how they are handled and translated to JSON messages.

We provide an implementation for an I2C-based capacitance lick detector which interested developers may wish to copy if support for novel I2C or SPI. Users with less development experience wishing to expand the hardware capabilities of behaviorMatecould also develop adapters which can be triggered on a TTL input/output. Additionally, more information about the JSON API and how messages are transmitted to the PC by the arduino is described in point (2) and the expanded online documentation.

a. Additionally, because it's unclear how easy to incorporate arbitrary hardware with behaviorMate, the "Intranet of things" approach seems to lose attraction. Since currently, the manuscript focuses mainly on a specific set of hardware designed for a specific type of experiment, it's not clear what are the advantages of implementing communication over a local network as opposed to the typical connections using USB.

As opposed to serial communication protocols as typical with USB, networking protocols seamlessly function based on asynchronous message passing. Messages may be routed internally (e.g. to a PCs localhost address, i.e. 0.0.0..0) or to a variety of external hardware (e.g. using IP addresses such as those in the range 192.168.1.2 - 192.168.1.254). Furthermore, network-based communication allows modules, such as VR, to be added easily. behavoirMate systems can be easily expanded using low-cost Ethernet switches and consume only a single network adapter on the PC (e.g. not limited by the number of physical USB ports). Furthermore, UDP message passing is implemented in almost all modern programming languages in a platform independent manner (meaning that the same software can run on OSX, Windows, and Linux). Lastly, as we have pointed out (Line 117) a variety of tools exist for inspecting network packets and debugging; meaning that it is possible to run behaviorMate with simulated hardware for testing and debugging.

The IOT nature of behaviorMate means there is no requirement for novel hardware to be implemented using an arduino, since any system capable of UDP communication can be configured. For example, VRMate is usually run on Odroid C4s, however one could easily create a system using Raspberry Pis or even additional PCs. behaviorMate is agnostic to the format of the UDP messages, but packaging any data in the JSON format for consistency would be encouraged. If a new hardware is a sensor that has input requiring it to be time stamped and logged then all that is needed is to add the IP address and port information to the ‘controllers’ list in a behaviorMate settings file. If more complex interactions are needed with novel hardware than a custom implementation of ContextList.java may be required (see response to R2.2). However, the provided UdpComms.java class could be used to easily send/receive messages from custom Context.java subclasses.

Solutions for highly customized hardware do require basic familiarity with object-oriented programming using the Java programming language. However, in our experience most behavioral experiments do not require these kinds of modifications. The majority of 1D navigation tasks, which behaviorMate is currently best suited to control, require touch/motion sensors, LEDs, speakers, or solenoid valves, which are easily controlled by the existing GPIO implementation. It is unlikely that custom subclasses would even be needed.

**Reviewer #3 (Public review):**
(1) While using UDP for data transmission can enhance speed, it is thought that it lacks reliability. Are there error-checking mechanisms in place to ensure reliable communication, given its criticality alongside speed?

The provided GPIO/behavior controller implementation sends acknowledgement packets in response to all incoming messages as well as start and stop messages for contexts and “valves”. In this way the UI can update to reflect both requested state changes as well as when they actually happen (although there is rarely a perceptible gap between these two states unless something is unplugged or not functioning). See Line 85 in the revised manuscript “acknowledgement packets are used to ensure reliable message delivery to and from connected hardware”.

(2) Considering this year's price policy changes in Unity, could this impact the system's operations?

VRMate is not affected by the recent changes in pricing structure of the Unity project.

The existing compiled VRMate software does not need to be regenerated to update VR scenes, or implement new task logic (since this is handled by the behaviorMate GUI). Therefore, the VRMate program is robust to any future pricing changes or other restructuring of the Unity program and does not rely on continued support of Unity. Additionally, while the solution presented in VRMate has many benefits, a developer could easily adapt any open-source VR Maze project to receive the UDP-based position updates from behaviorMate or develop their own novel VR solutions.

(3) Also, does the Arduino offer sufficient precision for ephys recording, particularly with a 10ms check?

Electrophysiology recording hardware typically has additional I/O channels which can provide assistance with tracking behavior/synchronization at a high resolution. While behaviorMate could still be used to trigger reward valves, either the ephys hardware or some additional high-speed DAQ would be recommended to maintain accurately with high-speed physiology data. behaviorMate could still be set up as normal to provide closed and open-loop task control at behaviorally relevant timescales alongside a DAQ circuit recording events at a consistent temporal resolution. While this would increase the relative cost of the individual recording setup, identical rigs for training animals could still be configured without the DAQ circuit avoiding unnecessary cost and complexity.

(4) Could you clarify the purpose of the Sync Pulse? In line 291, it suggests additional cues (potentially represented by the Sync Pulse) are needed to align the treadmill screens, which appear to be directed towards the Real-Time computer. Given that event alignment occurs in the GPIO, the connection of the Sync Pulse to the Real-Time Controller in Figure 1 seems confusing.

A number of methods exist for synchronizing recording devices like microscopes or electrophysiology recordings with behaviorMate’s time-stamped logs of actuators and sensors. For example, the GPIO circuit can be configured to send sync triggers, or receive timing signals as input. Alternatively a dedicated circuit could record frame start signals and relay them to the PC to be logged independently of the GPIO (enabling a high-resolution post-hoc alignment of the time stamps). The optimal method to use varies based on the needs of the experiment. Our setups have a dedicated BNC output and specification in the settings file that sends a TTL pulse at the start of an experiment in order to trigger 2p imaging setups (see line 224, specifically that this is a detail of “our” 2p imaging setup). We provide this information as it might be useful suggesting how to have both behavior and physiology data start recording at the same time. We do not intend this to be the only solution for alignment. Figure 1 indicates an “optional” circuit for capturing a high speed sync pulse and providing time stamps back to the real time PC. This is another option that might be useful for certain setups (or especially for establishing benchmarks between behavior and physiology recordings). In our setup event alignment does not exclusively occur on the GPIO.

a. Additionally, why is there a separate circuit for the treadmill that connects to the UI computer instead of the GPIO? It might be beneficial to elaborate on the rationale behind this decision in line 260.

Event alignment does not occur on the GPIO, separating concerns between position tracking and more general input/output features which improves performance and simplifies debugging. In this sense we maintain a single event loop on the Arduino, avoiding the need to either run multithreaded operations or rely extensively on interrupts which can cause unpredictable code execution (e.g. when multiple interrupts occur at the same time). Our position tracking circuit is therefore coupled to a separate,low-cost arduino mini which has the singular responsibility of position-tracking.

b. Moreover, should scenarios involving pupil and body camera recordings connect to the Analog input in the PCB or the real-time computer for optimal data handling and processing?

Pupil and body camera recordings would be independent data streams which can be recorded separately from behaviorMate. Aligning these forms of full motion video could require frame triggers which could be configured on the GPIO board using single TTL like outputs or by configuring a valve to be “pulsed” which is a provided type customization.

We also note that a more advanced developer could easily leverage camera signals to provide closed loop control by writing an independent module that sends UDP packets to behavoirMate. For example a separate computer vision based position tracking module could be written in any preferred language and use UDP messaging to send body tracking updates to the UI without editing any of the behaviorMate source code (and even used for updating 1D location).

(5) Given that all references, as far as I can see, come from the same lab, are there other labs capable of implementing this system at a similar optimal level?

To date two additional labs have published using behaviorMate, the Soltez and Henn labs (see revised lines 341-342). Since behaviorMate has only recently been published and made available open source, only external collaborators of the Losonczy lab have had access to the software and design files needed to do this. These collaborators did, however, set up their own behavioral setups in separate locations with minimal direct support from the authors–similar to what would be available to anyone seeking to set a behaviorMate system would find online on our github page or by posting to the message board.

**Recommendations for the authors:**

**Reviewer #1 (Recommendations For The Authors):**
(4) To provide additional context for the significance of this work, additional citations would be helpful to demonstrate a ubiquitous need for a system like behaviorMate. This was most needed in the paragraph from lines 46-65, specifically for each sentence after line 55, where the authors discuss existing variants on head-fixed behavioral paradigms. For instance, for the clause "but olfactory and auditory stimuli have also been utilized at regular virtual distance intervals to enrich the experience with more salient cues", suggested citations include Radvansky & Dombeck 2018 (DOI: 10.1038/s41467-018-03262-4), Fischler-Ruiz et al. 2021 (DOI: 10.1016/j.neuron.2021.09.055).

We thank the reviewer for the suggested missing citations and have updated the manuscript accordingly (see line 58).

(5) In addition, it would also be helpful to clarify behaviorMate's implementation in other laboratories. On line 304 the authors mention "other labs" but the following list of citations is almost exclusively from the Losonczy lab. Perhaps the citations just need to be split across the sentence for clarity? E.g. "has been validated by our experimental paradigms" (citation set 1) "and successfully implemented in other labs as well" (citation set 2).

We have split the citation set as suggested (see lines 338-342).

Minor Comments:(6) In the paragraph starting line 153 and in Fig. 2, please clarify what is meant by "trial" vs. "experiment". In many navigational tasks, "trial" refers to an individual lap in the environment, but here "trial" seems to refer to the whole behavioral session (i.e. synonymous with "experiment"?).

In our software implementation we had originally used “trial” to refer to an imaging session rather than experiment (and have made updates to start moving to the more conventional lexicon). To avoid confusion we have remove this use of “trial” throughout the manuscript and replaced with “experiment” whenever possible

(7) This is very minor, but in Figure 3 and 4, I don't believe the gavage needle is actually shown in the image. This is likely to avoid clutter but might be confusing to some readers, so it may be helpful to have a small inset diagram showing how the needle would be mounted.

We assessed the image both with and without the gavage needle and found the version in the original (without) to be easier to read and less cluttered and therefore maintained that version in the manuscript.

(8) In Figure 5 legend, please list n for mice and cells.

We have updated the Figure 5 legend to indicate that for panels C-G, n=6 mice (all mice were recorded in both VR and TM systems), 3253 cells in VR classified as significantly tuned place cells VR, and 6101 tuned cells in TM,

(9) Line 414: It is not necessary to tilt the entire animal and running wheel as long as the head-bar clamp and objective can rotate to align the imaging window with the objective's plane of focus. Perhaps the authors can just clarify the availability of this option if users have a microscope with a rotatable objective/scan head.

We have added the suggested caveat to the manuscript in order to clarify when the goniometers might be useful (see lines 281-288).

(10) Figure S1 and S2 could be referenced explicitly in the main text with their related main figures.

We have added explicit references to figures S1 and S2 in the relevant sections (see lines 443, 460 and 570)

(11) On line 532-533, is there a citation for "proximal visual cues and tactile cues (which are speculated to be more salient than visual cues)"?

We have added citations to both Knierim & Rao 2003 and Renaudineau et al. 2007 which discuss the differential impact of proximal vs distal cues during navigation as well as Sofroniew et al. 2014 which describe how mice navigate more naturally in a tactile VR setup as opposed to purely visual ones.

(12) There is a typo at the end of the Figure 2 legend, where it should say "Arduino Mini."

This typo has been fixed.

**Reviewer #2 (Recommendations For The Authors):**
(4) As mentioned in the public review: what is the major advantage of taking the IoT approaches as opposed to USB connections to the host computer, especially when behaviorMate relies on a central master computer regardless? The authors mentioned the readability of the JSON messages, making the system easier to debug. However, the flip side of that is the efficiency of data transmission. Although the bandwidth/latency is usually more than enough for transmitting data and commands for behavior devices, the efficiency may become a problem when neural recording devices (imaging or electrophysiology) need to be included in the system.

behaviorMate is not intended to do everything, and is limited to mainly controlling behavior and providing some synchronizing TTL style triggers. In this way the system can easily and inexpensively be replicated across multiple recording setups; particularly this is useful for constructing additional animal training setups. The system is very much sufficient for capturing behavioral inputs at relevant timescales (see the benchmarks in Figures 3 and 4 as well as the position correlated neural activity in Figures 5 and 6 for demonstration of this). Additional hardware might be needed to align the behaviorMate output with neural data for example a high-speed DAQ or input channels on electrophysiology recording setups could be utilized (if provided). As all recording setups are different the ideal solution would depend on details which are hard to anticipate. We do not mean to convey that the full neural data would be transmitted to the behaviorMate system (especially using the JSON/UDP communications that behaviorMate relies on).

(5) The author mentioned labView. A popular open-source alternative is bonsai (https://github.com/bonsai-rx/bonsai). Both include a graphical-based programming interface that allows the users to easily reconfigure the hardware system, which behaviorMate seems to lack. Additionally, autopilot (https://github.com/auto-pi-lot/autopilot) is a very relevant project that utilizes a local network for multiple behavior devices but focuses more on P2P communication and rigorously defines the API/schema/communication protocols for devices to be compatible. I think it's important to include a discussion on how behaviorMate compares to previous works like these, especially what new features behaviorMate introduces.

We believe that behaviorMate provides a more opinionated and complete solution than the projects mentioned. A wide variety of 1D navigational paradigms can be constructed in behaviorMate without the need to write any novel software. For example, bonsai is a “visual programming language” and would require experimenters to construct a custom implementation of each of their experiments. We have opted to use Java for the UI with distributed computations across modules in various languages. Given the IOT methodology it would be possible to use any number of programming languages or APIs; a large number of design decisions were made when building the project and we have opted to not include this level of detail in the manuscript in order to maintain readability. We strongly believe in using non-proprietary and open source projects, when possible, which is why the comparison with LabView based solutions was included in the introduction. Also, we have added a reference to the autopilot reference to the section of the introduction where this is discussed.

(6) One of the reasons labView/bonsai are popular is they are inherently parallel and can simultaneously respond to events from different hardware sources. While the JSON events in behaviorMate are asynchronous in nature, the handling of those events seems to happen only in a main event loop coupled with GUI, which is sequential by nature. Is there any multi-threading/multi-processing capability of behaviorMate? If so it's an important feature to highlight. If not I think it's important to discuss the potential limitation of the current implementation.

IOT solutions are inherently concurrent since the computation is distributed. Additional parallelism could be added by further distributing concerns between additional independent modules running on independent hardware. The UI has an eventloop which aggregates inputs and then updates contexts based on the current state of those inputs sequentially. This sort of a “snapshot” of the current state is necessary to reason about when the start certain contexts based on their settings and applied decorators. While the behaviorMate UI uses multithreading libraries in Java to be more performant in certain cases, the degree to which this represents true vs “virtual” concurrency would depend on the individual PC architecture it is run on and how the operating system allocates resources. For this reason, we have argued in the manuscript that behaviorMate is sufficient for controlling experiments at behaviorally relevant timescales, and have presented both benchmarks and discussed different synchronization approaches and permit users to determine if this is sufficient for their needs.

(7) The context list is an interesting and innovative approach to abstract behavior contingencies into a data structure, but it's not currently discussed in depth. I think it's worth highlighting how the context list can be used to cover a wide range of common behavior experimental contingencies with detailed examples (line 185 might be a good example to give). It's also important to discuss the limitation, as currently the context lists seem to only support contingencies based purely on space and time, without support for more complicated behavior metrics (e.g. deliver reward only after X% correct).

To access more complex behavior metrics during runtime, custom context list decorators would need to be implemented. While this is less common in the sort of 1D navigational behaviors the project was originally designed to control, adding novel decorators is a simple process that only requires basic object oriented programming knowledge. As discussed we are also implementing a plugin-architecture in the JavaFX update to streamline these types of additions.

Minor Comments:(8) In line 202, the author suggests that a single TTL pulse is sent to mark the start of a recording session, and this is used to synchronize behavior data with imaging data later. In other words, there are no synchronization signals for every single sample/frame. This approach either assumes the behavior recording and imaging are running on the same clock or assumes evenly distributed recording samples over the whole recording period. Is this the case? If so, please include a discussion on limitations and alternative approaches supported by behaviorMate. If not, please clarify how exactly synchronization is done with one TTL pulse.

While the TTL pulse triggers the start of neural data in our setups, various options exist for controlling for the described clock drift across experiments and the appropriate one depends on the type of recordings made, frame rate duration of recording etc. Therefore behaviorMate leaves open many options for synchronization at different time scales (e.g. the adding a frame-sync circuit as shown in Figure 1 or sending TTL pulses to the same DAQ recording electrophysiology data). Expanded consideration of different synchronization methods has been included in the manuscript (see lines 224-238).

(9) Is the computer vision-based calibration included as part of the GUI functionality? Please clarify. If it is part of the GUI, it's worth highlighting as a very useful feature.

The computer vision-based benchmarking is not included in the GUI. It is in the form of a script made specifically for this paper. However for treadmill-based experiments behaviorMate has other calibration tools built into it (see line 301-303).

(10) I went through the source code of the Arduino firmware, and it seems most "open X for Y duration" functions are implemented using the delay function. If this is indeed the case, it's generally a bad idea since delay completely pauses the execution and any events happening during the delay period may be missed. As an alternative, please consider approaches comparing timestamps or using interrupts.

We have avoided the use of interrupts on the GPIO due to the potential for unpredictable code execution. There is a delay which is only just executed if the duration is 10 ms or less as we cannot guarantee precision of the arduino eventloop cycling faster than this. Durations longer than 10 ms would be time stamped and non-blocking. We have adjusted this MAX_WAIT to be specified as a macro so it can be more easily adjusted (or set to 0).

(11) Figure 3 B, C, D, and Figure 4 D, E suffer from noticeable low resolution.

We have converted Figure 3B, C, D and 4C, D, E to vector graphics in order to improve the resolution.

(12) Figure 4C is missing, which is an important figure.

This figure appeared when we rendered and submitted the manuscript. We apologize if the figure was generated such that it did not load properly in all pdf viewers. The panel appears correctly in the online eLife version of the manuscript. Additionally, we have checked the revision in Preview on Mac OS as well as Adobe Acrobat and the built-in viewer in Chrome and all figure panels appear in each so we hope this issue has been resolved.

(13) There are thin white grid lines on all heatmaps. I don't think they are necessary.

The grid lines have been removed from the heatmaps as suggested.

(14) Line 562 "sometimes devices directly communicate with each other for performance reasons", I didn't find any elaboration on the P2P communication in the main text. This is potentially worth highlighting as it's one of the advantages of taking the IoT approaches.

In our implementation it was not necessary to rely on P2P communication beyond what is indicated in Figure 1. The direct communication referred to in line 562 is meant only to refer to the examples expanded on in the rest of the paragraph i.e. the behavior controller may signal the microscope directly using a TTL signal without looping back to the UI. As necessary users could implement UDP message passing between devices, but this is outside the scope of what we present in the manuscript.

(15) Line 147 "Notably, due to the systems modular architecture, different UIs could be implemented in any programming language and swapped in without impacting the rest of the system.", this claim feels unsupported without a detailed discussion of how new code can be incorporated in the GUI (plugin system).

This comment refers to the idea of implementing “different UIs”. This would entail users desiring to take advantage of the JSON messaging API and the proposed electronics while fully implementing their own interface. In order to facilitate this option we have improved documentation of the messaging API posted in the README file accompanying the arduino source code. We have added reference to the supplemental materials where readers can find a link to the JSON API implementation to clarify this point.

Additionally, while a plugin system is available in the JavaFX version of behaviorMate, this project is currently under development and will update the online documentation as this project matures, but is unrelated to the intended claim about completely swapping out the UI.

**Reviewer #3 (Recommendations For The Authors):**
(6) Figure 1 - the terminology for each item is slightly different in the text and the figure. I think making the exact match can make it easier for the reader.- Real-time computer (figure) vs real-time controller (ln88).

The manuscript was adjusted to match figure terminology.

- The position controller (ln565) - position tracking (Figure).

We have updated Figure 1 to highlight that the position controller does the position tracking.

- Maybe add a Behavior Controller next to the GPIO box in Figure 1.

We updated Figure 1 to highlight that the Behavior Controller performs the GPIO responsibility such that "Behavior Controller" and "GPIO circuit" may be used interchangeably.

- Position tracking (fig) and position controller (subtitle - ln209).

We updated Figure 1 to highlight that the position controller does the position tracking.

- Sync Pulse is not explained in the text.

The caption for Figure 1 has been updated to better explain the Sync pulse and additional systems boxes

(7) For Figure 3B/C: What is the number of data points? It would be nice to see the real population, possibly using a swarm plot instead of box plots. How likely are these outliers to occur?

In order to better characterize the distributions presented in our benchmarking data we have added mean and standard deviation information the plots 3 and 4. For Figure 3B: 0.0025 +/- 0.1128, Figure 3C: 12.9749 +/- 7.6581, Figure 4C: 66.0500 +/- 15.6994, Figure 4E: 4.1258 +/- 3.2558.